# Direct single-molecule detection and super-resolution imaging with a low-cost portable smartphone-based microscope

Morgane Loretan[1], Mariano Barella [1,2] ✉, Nathan Fuchs [1], Samet Kocabey [2,3], Karol Kołątaj [1,2], Fernando D. Stefani [4] ✉ & Guillermo P. Acuna [1,2] ✉

Detecting single molecules, which represents the ultimate level of sensitivity, is typically achieved with research-grade equipment. Here we present a low-cost, portable smartphone-based fluorescence microscope capable of detecting single-molecule fluorescence directly, i.e., without the need for any signal amplification. The setup leverages the image sensors and data handling capacity of mass-produced smartphones, it is adaptable to different smartphones and capable of detecting single molecules across the visible spectral range. We showcase this capability through single-molecule measurements on DNA origami models and super-resolution microscopy of cells by single-molecule localization microscopy. Last, we illustrate its potential as a point-of-care (POC) device by implementing a single-molecule bioassay for RNA detection. This development paves the way for innovative applications of massively distributed or personalized assays with single-molecule sensitivity in various fields such as digital bioassays, POC diagnostics, field expeditions, STEM outreach, and life science education.

Presently, the global proliferation of smartphones is estimated to be around 7 billion units[1]. This vast scale of production continually propels the integration of technological advancements at a cost-efficiency unparalleled by other sectors. The unrivaled portability, compactness, and worldwide accessibility, coupled with high-performance image sensors, robust computing power, and connectivity, have catalyzed the development of specialized smartphone-based setups for Point-Of-Care (POC) and personal biomedical applications[2]. These setups leverage the advanced camera technology of modern smartphones[3] for fluorescence-based clinical diagnostics[4], quantification of immunoassays[5,6], detection of bacteria[7], cancer cytology[8], fresh tissue imaging[9], and environmentally significant measurements such as lead and microplastics quantification[10,11].

Since the pioneering work by the Ozcan group a decade ago[12], which demonstrated the detection of micrometer-long stained double-stranded DNA and fluorescent beads, the sensitivity of smartphone-based fluorescence microscopes has been continually improved. Naturally, there has been a quest to achieve single-molecule sensitivity to unlock all the advantages of sub-ensemble determinations for personal or POC assays. For instance, it would be possible to quantify target molecules under ultra-low analyte concentrations by implementing digital bioassays[13,14] or to conduct super-resolution fluorescence imaging[15–17] with a portable and cost-effective instrument. While other low-cost solutions for single-molecule detection have been developed[16,17], the use of smartphone-based setups holds unique potential due to their portability and intrinsic connectivity. To the best

[1]Department of Physics, Faculty of Science and Medicine, University of Fribourg, Fribourg, Switzerland. [2]Swiss National Center for Competence in Research (NCCR) Bio-inspired Materials, University of Fribourg, Fribourg, Switzerland. [3]Department of Oncology, Microbiology and Immunology, Faculty of Science and Medicine, University of Fribourg, Fribourg, Switzerland. [4]Centro de Investigaciones en Bionanociencias (CIBION), Consejo Nacional de Investigaciones Científicas y Técnicas (CONICET) and Departamento de Física, Facultad de Ciencias Exactas y Naturales, Universidad de Buenos Aires, Ciudad Autónoma de Buenos Aires, Argentina. ✉e-mail: mariano.barella@unifr.ch; fernando.stefani@df.uba.ar; guillermo.acuna@unifr.ch

of our knowledge, the highest sensitivity reported for a smartphone-based microscope was 10 fluorophores, and this required highly specialized hardware like research-grade optical components and lasers, a particular smartphone model with a monochromatic camera, and the measurements had to be performed on an optical table[18]. Until now, detecting single molecules in smartphone-based setups was only possible using physical or chemical amplification mechanisms, which are hardly compatible with POC or personal applications because they involve rather complex procedures and require trained personnel. For instance, single-molecule fluorescence was detected using a smartphone microscope by strongly enhancing the fluorescence signal with DNA origami-based optical antennas[19]. However, assembling these sophisticated nanostructures, as well as handling them for single-molecule measurements, must be done following precise protocols. Moreover, the amplification of the fluorescence signal exhibits an intrinsic dispersion that hinders the implementation in quantitative sensing applications[19]. This variability is further increased by slight variations in sample preparation. Alternatively, chemical amplification approaches have been implemented to increase the number of copies of the analyte. For instance, several works have combined digital polymerase chain reaction (dPCR) with smartphone-based fluorescence microscopy[20]. While this approach enables the detection of single gene expressions in a low number of HepG2 cells[21], it requires precise temperature control to perform the necessary heating and cooling cycles, as well as the timely exchange of different reagents in the sample.

Here, we present a portable and inexpensive smartphone-based fluorescence microscope capable of detecting single molecules using three commercially available smartphones from different manufacturers. The microscope enables direct detection of single-molecule fluorescence, without the need to amplify the fluorescence signal or the number of emitting molecules. These features facilitate the widespread application of single-molecule assays and measurements. Its performance was first demonstrated by detecting single-molecule intensity fluctuations in DNA origami model systems with a favorable signal-to-noise ratio of 3.3. Furthermore, we successfully leveraged the microscope's single-molecule sensitivity to achieve super-resolution microscopy. By implementing a type of Single-Molecule Localization Microscopy (SMLM) called DNA-PAINT (Points Accumulation for Imaging in Nanoscale Topography), we were able to super-resolve both DNA origami structures and microtubule networks of cells. The acquired images exhibited a localization precision of 84 nm, translating to a 6.6-fold enhancement in resolution. Finally, we proved the potential of the smartphone-based microscope for POC applications and sensing by detecting Ebola RNA fragments through DNA-PAINT measurements on a DNA origami-based digital bioassay. In summary, this work represents a significant advancement towards making single-molecule fluorescence assays and methods widely accessible, with potential applications in various fields, from point-of-care quantitative sensing devices to nanoscale imaging for personalized diagnostics.

## Results

### The smartphone-based microscope

Figure 1a includes a photograph of the smartphone-based fluorescence microscope. The microscope weighs 1.2 kg, and its dimensions are $11 \times 22 \times 12$ cm, smaller than a standard shoe box. It is a stand-alone unit that includes a laser, a power source, and the necessary optical components in a modular design conceived to optimize sensitivity, portability, affordability, and practicality (additional photographs can be found in Supplementary Note 1). It can be operated following simple user instructions on any conventional table, or even on the floor. Using the microscope requires minimal training, achievable through instructions, a brief tutorial, or a video demonstration. The cost of the microscope lies under 350 € (see list of components and respective prices in Methods and Supplementary Note 2). Finally, in

contrast to other designs that focus on a particular model, our microscope is flexible enough to host virtually any smartphone with a camera. To illustrate this versatility, we benchmarked its performance with three different models from leading global smartphone producers, Apple, Samsung, and Huawei.

Figure 1b depicts a sketch of the optical paths introduced to enhance the microscope's sensitivity by minimizing the background signal. Fluorescence excitation is achieved with a laser beam that passes through a focusing lens (FL) and reaches a half-ball lens that acts as a prism for total internal reflection (TIR) illumination. The half-ball is glued to a glass slide (prism holder) using an optical adhesive. Immersion oil is applied between the prism holder and the sample substrate to match the refractive indices and complete the TIR configuration. This configuration differs from previous smartphone-based microscopes that used waveguided LEDs as light sources, which, in contrast to lasers, are less radiant and spectrally broader[18,19,22–28]. The light emitted by the sample is collected by an inexpensive, low numerical aperture (NA) air objective (Obj), spectrally selected with an emission filter (EF), and focused onto the smartphone CMOS sensor by its camera lens, acting as a tube lens (TL).

Figure 1b also depicts the modular design of the smartphone-based microscope. The setup is composed of four parts: the protective black case, the laser stage (purple), the objective stage (yellow), and the sample stage (pink). The protective black case comes with a top half-cover that shields the user from laser radiation and allows the smartphone to be placed using two slip-resistant silicone supports that can be freely moved. This simple configuration makes the setup compatible with smartphones of different sizes and camera positions (Fig. 1c). The protective case also hosts the laser control electronics and the battery, as shown in Fig. 1d. During transportation, it is used to store all the optical components, immersion oil, and samples (see Supplementary Note 1). Inside, four magnets allow the laser stage to be easily removed and repositioned without needing beam realignment (Fig. 1e). This gives the setup flexibility for selecting the excitation wavelength, as different lasers can be easily interchanged. The removable sample and objective stages fit into a supporting frame (SF, Fig. 1e).

The laser stage shown in Fig. 1f comprises the laser module with the focusing lens, a heatsink, and alignment screws. A cooling fan at the end of the heatsink is optionally mounted for high-power lasers. The fan is turned off or removed to reduce possible vibrations when high localization precision is desired, like during SMLM measurements. The laser stage features three degrees of freedom that allow translation over the microscope horizontal plane ($xy$) and fine-tuning the incidence angle ($\theta$) to achieve highly inclined and laminated optical sheet (HILO) or TIR illumination.

The objective stage shown in Fig. 1g has been conceived to focus on the sample plane ($z$-axis) and to align the objective with the illuminated area of the sample ($xy$ plane). These degrees of freedom, controlled by alignment screws, offer flexibility when using different sample substrates without the need to realign the laser. The objective holder contains the low-cost objective and the emission filter. The emission filter is installed in a lateral slot, enabling easy exchange without touching the smartphone. The objective can also be exchanged if required.

To be able to move the sample over $x$ and $y$ directions, we designed a sample stage (Fig. 1h) that houses two different moving elements operated by two screws. On the moving stage, the sample holder includes magnets to secure the sample with a top holder. The prism holder is glued to the stage on the bridge, right below the sample. A white light-emitting diode (LED) is incorporated in a cavity inside the bridge, next to the prism, that allows image pre-focusing, sample positioning, and bright-field imaging. An LED blocker is used to prevent the laser from exciting the LED undesirably. Similarly, a manual shutter (Beam blocker) is included to obstruct the laser beam from

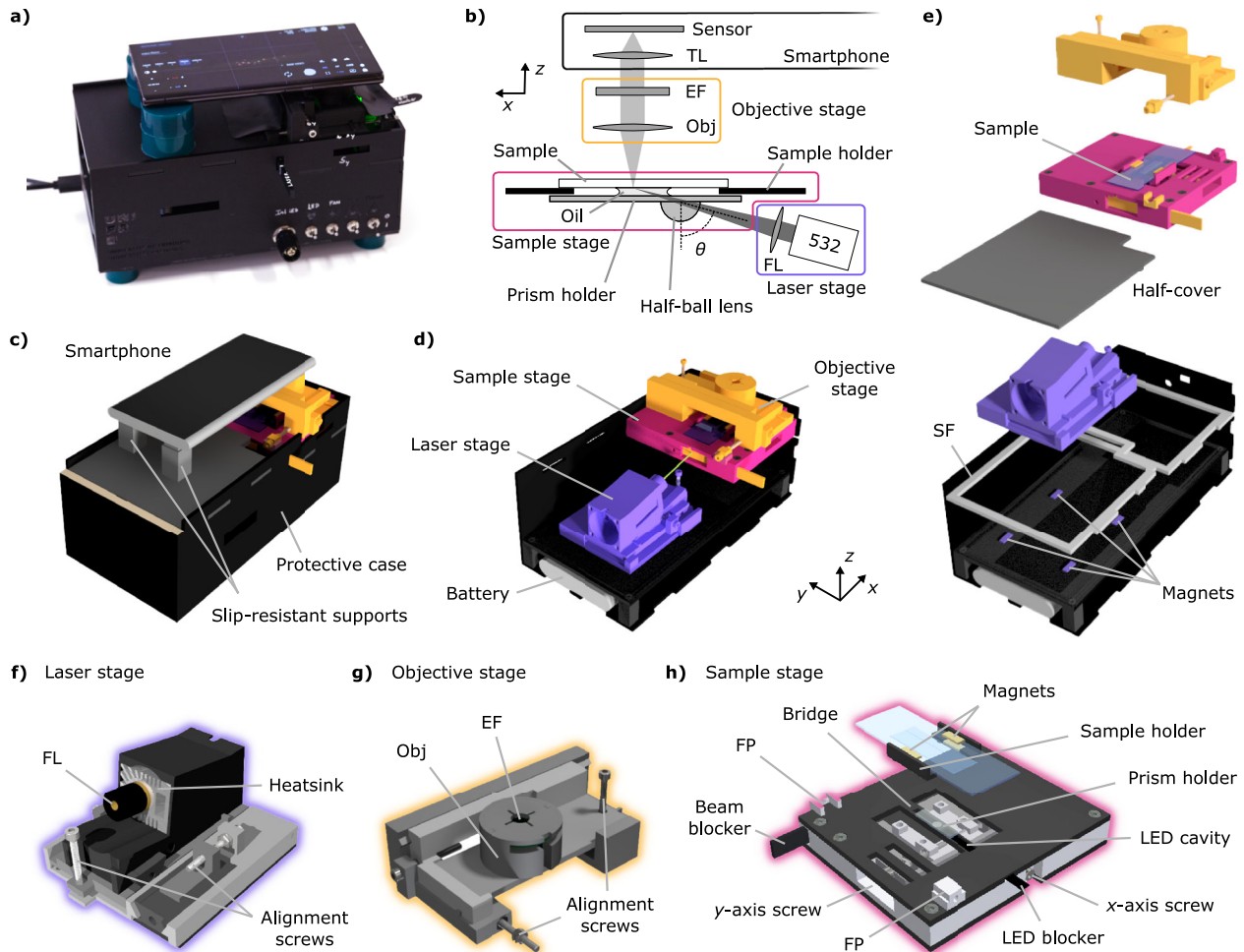

**Fig. 1 | Smartphone-based fluorescence microscope. a** Photograph of the setup with a Samsung Galaxy S22 Ultra on top. **b** Optical path sketch of the smartphone-based microscope. Each color frame encloses the optical components of the objective stage (yellow), sample stage (pink), laser stage (purple), and the smartphone camera (black). **c** A rendered model of the smartphone-based microscope ready for use. **d** Inside view with the laser, sample, and objective stages. **e** Exploded view depicting the supporting frame (SF) and the laser stage magnets. **f**– **h** Details of the three stages of the microscope with their main components. Obj: Objective. TL: tube lens. FL: focusing lens. EF: emission filter. FP: fixation point.

reaching the sample. Finally, the sample stage features two small fixation points (FP) that guide and fix the objective stage position.

### Direct single-molecule detection with the smartphone-based microscope

To test the ability of our portable microscope to detect the emission of a single fluorescent molecule, we used fluorescence standards based on DNA origami structures[29]. The design consists of a $60 \times 52$ nm$^2$ 2-layer sheet origami (2LS) with an ATTO 542 dye at the center, and an ATTO 647N dye 22 nm to the side (Fig. 2a). Six biotins are included on the bottom side for surface binding. Transmission Electron Microscopy (TEM) images of the 2LS are shown in Supplementary Note 3.

A sample with DNA origami immobilized on a quartz substrate at a density suited for single-molecule measurements was prepared. We first imaged the sample using a custom-built high-end widefield fluorescence microscope to detect the ATTO 647N dyes (see Methods). Exciting at 640 nm, we recorded a series of images over time (100 ms exposure time), where the single-step photobleaching of single ATTO 647N molecules could be observed (see Supplementary Notes 4 and 19 for experimental details). Figure 2b shows an exemplary image (10-frame average, total acquisition time 1 s), where single ATTO 647N molecules from several DNA origami structures can be clearly detected as a diffraction-limited spot corresponding to the point spread

function (PSF) of the system. Next, the sample was moved onto the smartphone-based microscope placed over a standard desk to detect the ATTO 542 molecules excited at 532 nm. The sample position was adjusted to image the region observed previously with the high-end microscope. A Samsung Galaxy S22 Ultra smartphone was used to record a video with the MotionCam Pro app without compression (raw data mode), with an exposure time of 250 ms. Figure 2c shows a fluorescence image obtained from averaging 16 frames (total acquisition time 4 s – see Supplementary Note 19 for further experimental parameters). A first visual inspection already indicates that single ATTO 542 molecules can be detected with the smartphone-based microscope, albeit with a bigger PSF due to the lower NA of the employed objective. An overlay of the images allowed us to correlate the presence of single ATTO 542 and ATTO 647N molecules incorporated in the same 2LS DNA origami (Fig. 2f) within the same diffraction-limited spot. We found that 89% of the fluorescent spots observed on the smartphone-based microscope correspond to origami nanostructures with a single ATTO 542 and a single ATTO 647N. The remaining spots belong to either origami nanostructures labeled with only a single ATTO 542 or unidentified fluorescence sources. These results are consistent with typical reported yields for the incorporation of single-stranded DNA sequences labeled with a single dye during DNA origami self-assembly (see further

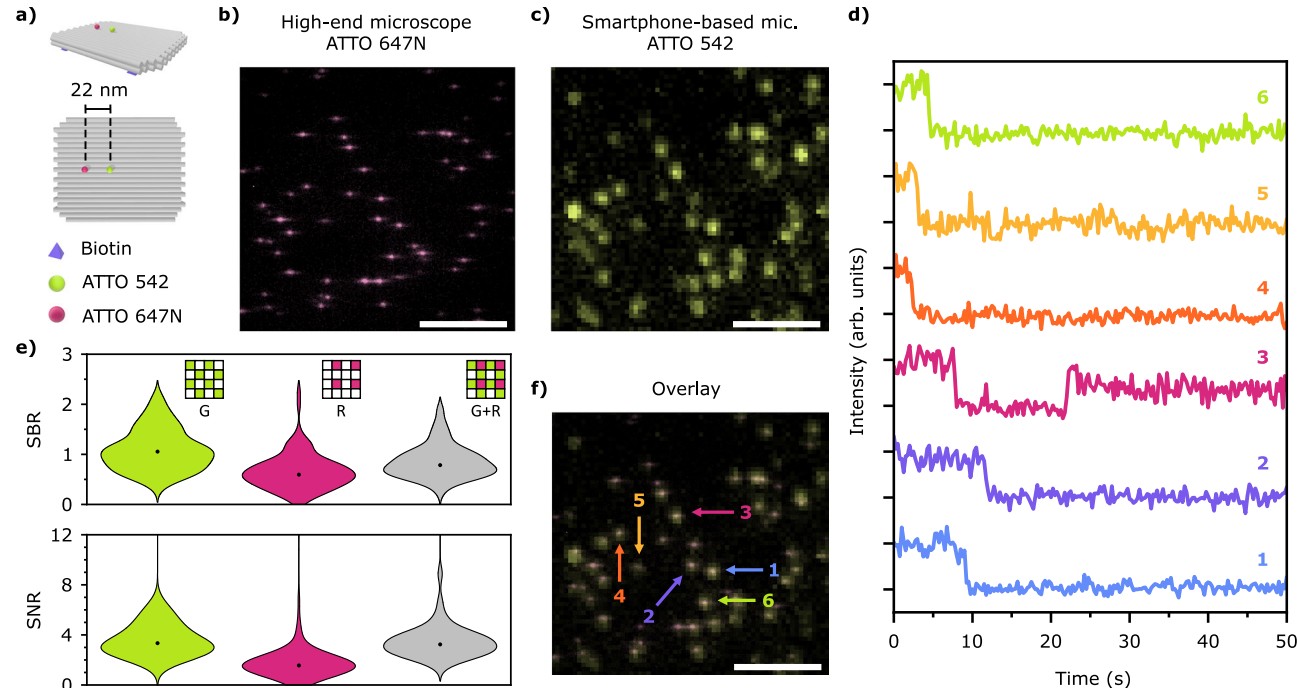

**Fig. 2 | Direct single-molecule detection with the smartphone-based microscope. a** 2LS DNA origami nanostructure was used to detect single molecules with an ATTO 542 at the center and an ATTO 647N spaced 22 nm apart. Fluorescence images of the 2LS origami sample were recorded over the same area with (**b**) the high-end microscope detecting only the ATTO 647N and (**c**) the smartphone-based microscope detecting only the ATTO 542 with a Samsung Galaxy S22 Ultra. The effective exposure times were 1 s and 4 s for (**b**) and (**c**), respectively. In total, four images were taken with both microscopes. Scale bar: 10 μm. **d** Intensity traces vs. time of several ATTO 542 single molecules. The signal after photobleaching represents the background level. No time averaging was performed. Each trace corresponds to the 2LS nanostructure indicated in the overlayed image shown in (**f**). **e** SBR, or Weber contrast, and SNR violin plots of 104 molecules acquired with the smartphone-based microscope using the green (G) channel, the red (R) channel, and the sum of both channels, green and red (G + R). Insets: Bayer filter pixel selection. **f** Overlay of images (**b**) and (**c**). Arrows and numbers indicate intensity traces plotted in (**d**). Scale bar: 10 μm. Source data are provided as a Source Data file.

details in Supplementary Note 5). This already provides solid evidence that our microscope can detect single ATTO 542 molecules. Further confirmation was obtained from fluorescence transients taken with the smartphone. Six exemplary intensity transient traces of single ATTO 542 molecules are shown in Fig. 2d, with their corresponding positions highlighted in Fig. 2f (numbered arrows). Binary blinking and single-step photobleaching events are clearly visible, which constitute unambiguous signatures of the photophysics of single fluorophores.

Finally, in order to quantify the microscope´s sensitivity for single-molecule detection, we estimated both the signal-to-background ratio (SBR), sometimes also referred to as Weber contrast, and the signal-to-noise ratio (SNR). Since data was acquired in raw mode, we have access to the three Bayer filter channels of the camera sensor (red, green, and blue). This enables choosing the most convenient combination of channels to maximize the SBR and the SNR, depending on the target fluorophore. Figure 2e shows the SBR and the SNR distributions for 104 ATTO 542 molecules detected on the green, red, and green + red channels. The blue channel was not considered, as the long-pass emission filter used had a cut-off wavelength of 550 nm. It is worth mentioning that the SBR was estimated from one frame and, therefore, quantifies the extent to which a single molecule can be detected from an image, whereas the SNR was extracted from the fluorescence transients (see further details in Methods). Median values of SBR and SNR were higher for the green channel alone ($SBR_G = 1.1$, $SNR_G = 3.3$) than for the red channel alone ($SBR_R = 0.60$, $SNR_R = 1.6$) or the sum of green and red channels ($SBR_{G+R} = 0.78$, $SNR_{G+R} = 3.2$). Thus, all single-molecule data shown for ATTO 542 were analyzed from the green channel of the smartphone sensors.

To assess the smartphone-based microscope's performance relative to the high-end microscope, SBR and SNR were estimated for the same fluorophore, ATTO 542, under similar experimental conditions (see Supplementary Note 7 and 19), both on glass and quartz slides. The median values for the high-end microscope were $SBR_{high-end} = 0.67$ (2.25) and $SNR_{high-end} = 9.9$ (16.8) on quartz (glass). Notably, on quartz slides, the SBR achieved with the high-end microscope is around 40% lower than the one obtained with the smartphone-based microscope, due to the refractive index mismatch between the high-end objective and the quartz substrate. Conversely, the difference between the SNR values on quartz slides might arise from the low-noise research-grade CMOS camera of the high-end microscope, which reduces the standard deviation of the fluorescence signal, increasing the SNR (see Methods). On glass slides, as expected, both the SBR and SNR recorded on the high-end microscope are higher than the values obtained with the smartphone-based counterpart.

Analogous measurements were performed with an iPhone 14 Pro and a Huawei P20 Pro (see Supplementary Note 6). The smartphone-based setup can detect single-molecule fluorescence with any of those smartphones. We note that the cameras of all employed smartphones are above average but do not include the latest developments. The Huawei P20 Pro was released in March 2018, the Samsung Galaxy S22 Ultra in February 2022, and the iPhone 14 Pro in September 2022. Given the current rapid evolution of smartphone technologies, it is expected that in the near future the great majority of smartphones will count with image acquisition systems delivering a performance comparable to the ones used in this work. Three camera parameters are crucial for single-molecule detection: the aperture, focal length, and sensor pixel size. A higher aperture (low f-number) is convenient as it

provides higher photon collection efficiency. The smartphone camera's focal length and objective lens determine the system's magnification, which in turn defines the size of the single-molecule signals projected on the camera sensor. For optimum sampling of the single-molecule signals, this should be distributed in about $3 \times 3$ sensor pixels[30,31]. This condition limited the choices of suitable cameras for our experiments. In addition, we wanted to have access to RAW data images in order to analyze the data with our algorithms. Considering all these factors, we performed single-molecule experiments with the best-suited camera of each smartphone, namely the wide color camera of the Huawei P20 Pro, the telephoto camera of the Samsung Galaxy S22 Ultra, and the telephoto camera of the iPhone 14 Pro. Further details about the characteristics of each camera are provided in Supplementary Table 2.

## Super-resolution benchmark with DNA origami models

Direct single-molecule fluorescence detection is a prerequisite for imaging beyond the diffraction limit with SMLM. Thus, among other exciting possibilities, our microscope has the potential to perform super-resolution microscopy using a mass-consumption smartphone. To demonstrate this, we used DNA origami as a nanoscopic ruler for DNA-PAINT[32,33]. The DNA origami nanostructure was designed based on an 8-helix bundle (8HB) with two docking sites separated by 256 nm, each composed of 11 or 12 docking strands, and biotins for binding to the surface, as depicted in Fig. 3a. TEM images of several 8HBs are shown in Supplementary Note 3.

Samples were prepared with 8HB DNA-origami dispersed on a quartz substrate at densities suited for their individual observation through DNA-PAINT. We used a fluorogenic imager strand with a Cy3B fluorophore on one of its ends and a BHQ-2 fluorescence quencher on the other to reduce the background level[34]. The imager had 12 complementary bases with the docking strands, delivering an average binding time of about 1 s. Gold nanoparticles were used as fiducial markers for drift correction (further details in Methods and Supplementary Note 9). Single-molecule blinking videos on these samples were acquired with both the high-end microscope (100 ms exposure time – 18,000 frames) and the smartphone-based setup with the Samsung Galaxy S22 Ultra (250 ms exposure time – 40,785 frames), and subsequently analyzed with Picasso software[35] to reconstruct super-resolution images from single-molecule localizations. The detection of single molecules with the smartphone-based setup works at rather low levels of SNR. We found it advantageous to perform a 3-frame average of the video before analysis with Picasso. This procedure improved the symmetry of the detected single-molecule signals, leading to more precise localizations (see Supplementary Note 11), but required the acquisition of $3\times$ more frames. This, in combination with the $2.5\times$ longer frame time, made the total acquisition times with the smartphone-based setup $7.5\times$ longer. Also, processing the acquired data by the smartphone to be exported in a useful format for further SMLM localization took significant time (2.2 h for a 40,000-frame video of 3.5 MB/frame).

Figures 3b and c show super-resolved images of DNA origami on the same region of a sample obtained from measurements on the smartphone-based setup and the high-end microscope, respectively. Remarkably, the smartphone-based setup delivers super-resolution images in excellent agreement with the high-end microscope measurements. A one-to-one comparison of individual images exhibits docking sites for DNA-PAINT with matching orientations and separation distances, as seen in the ten examples of Fig. 3d. A quantitative comparison of the separation between docking sites was made by computing the average 2D histograms of localizations of those ten nanostructures (Fig. 3e). Figure 3f displays the histograms of localizations along the main axis of the average super-resolved nanostructures (Fig. 3e). Each histogram was satisfactorily fitted with the sum of two Gaussian functions with, in principle, different

amplitudes and positions, but identical standard deviations (width). This allowed us to determine the average distances between docking sites as the distance between the Gaussians' positions, which were 260 nm and 248 nm for the measurements with the smartphone-based microscope and the high-end microscope, respectively. Both distances are in good agreement with the DNA origami design.

A measure of the average localization precision can be estimated as the standard deviation of the Gaussian functions ($\sigma$), which were 84 nm for the smartphone-based microscope and 24 nm for the high-end microscope. An estimate of the achieved resolution is provided by the respective full-width at half maximum (FWHM) values of 197 nm and 56 nm (FWHM = $2.35\sigma$), which are well below the FWHM of the experimentally determined PSFs: $(1.3 \pm 0.3)$ µm for the smartphone-based setup and $(0.27 \pm 0.01)$ µm for the high-end microscope (see Fig. 3g and Supplementary Note 11). The ratio of the FWHMs of the PSF and the super-resolved docking site of the DNA origami provides an estimation of the increase in resolution. These measurements retrieved a resolution increase of 6.6 times for the smartphone-based microscope and 4.8 times for the research-grade microscope. The higher resolution gain observed with the smartphone-based setup is attributable to the greater influence of the DNA-PAINT docking site size on the high-end microscope measurements. Specifically, the 28 nm docking site size is comparable to the 24 nm localization precision of the high-end microscope, whereas it is only one-third of the 84 nm localization precision achieved with the smartphone-based setup. This difference highlights the remarkable performance of the smartphone-based microscope, particularly given that the irradiance on the sample was lower compared to the high-end microscope (see Supplementary Note 19). As an additional characterization, we analyzed the achieved localization precision using the method called NeNa[36], based on first-neighbor distances on consecutive frames, on the super-resolved images. Running NeNa on the dataset acquired with the smartphone-based microscope yields a localization precision of 119 nm, a value larger than the standard deviation obtained from the Gaussian fit to the localization histograms (Fig. 3f). This is likely because NeNa evaluates all localizations in the area analyzed, including non-specific binding events outside the docking sites of the origami, which can contribute to enlarging the average first neighbor distance (see Supplementary Note 10 for further details on localization precision estimation).

## Super-resolution cell imaging with the smartphone-based microscope

To definitively showcase the versatility of the smartphone-based single-molecule detection platform, we undertook super-resolution imaging of the microtubule network of U2OS cells on standard glass coverslips and using the smartphone-based setup on a common desk table. α-tubulin was immunolabeled with primary-secondary antibodies for DNA-PAINT as sketched in Fig. 4a (see Methods for further details). We used Cy3B-labeled imager strands at a concentration of 500 pM, photostabilized in 1.5×PPC. Figure 4b shows a composite super-resolved / diffraction-limited fluorescence image of the microtubule network obtained with the smartphone setup (see Supplementary Note 12 for another microtubule network example). These results provide compelling evidence that the smartphone-based setup can achieve super-resolution imaging using SMLM. Despite the long recording times, the observed drift with the smartphone-based microscope is manageable even when used on a standard desktop (~3.5 µm in one hour, see Supplementary Note 9 for further details). Later, we again compared its performance to the high-end single-molecule microscope. Figure 4c and d present super-resolved images of the same region of the microtubule network, respectively obtained from videos acquired with smartphone-based and high-end microscopes. The achieved resolution was estimated via decorrelation

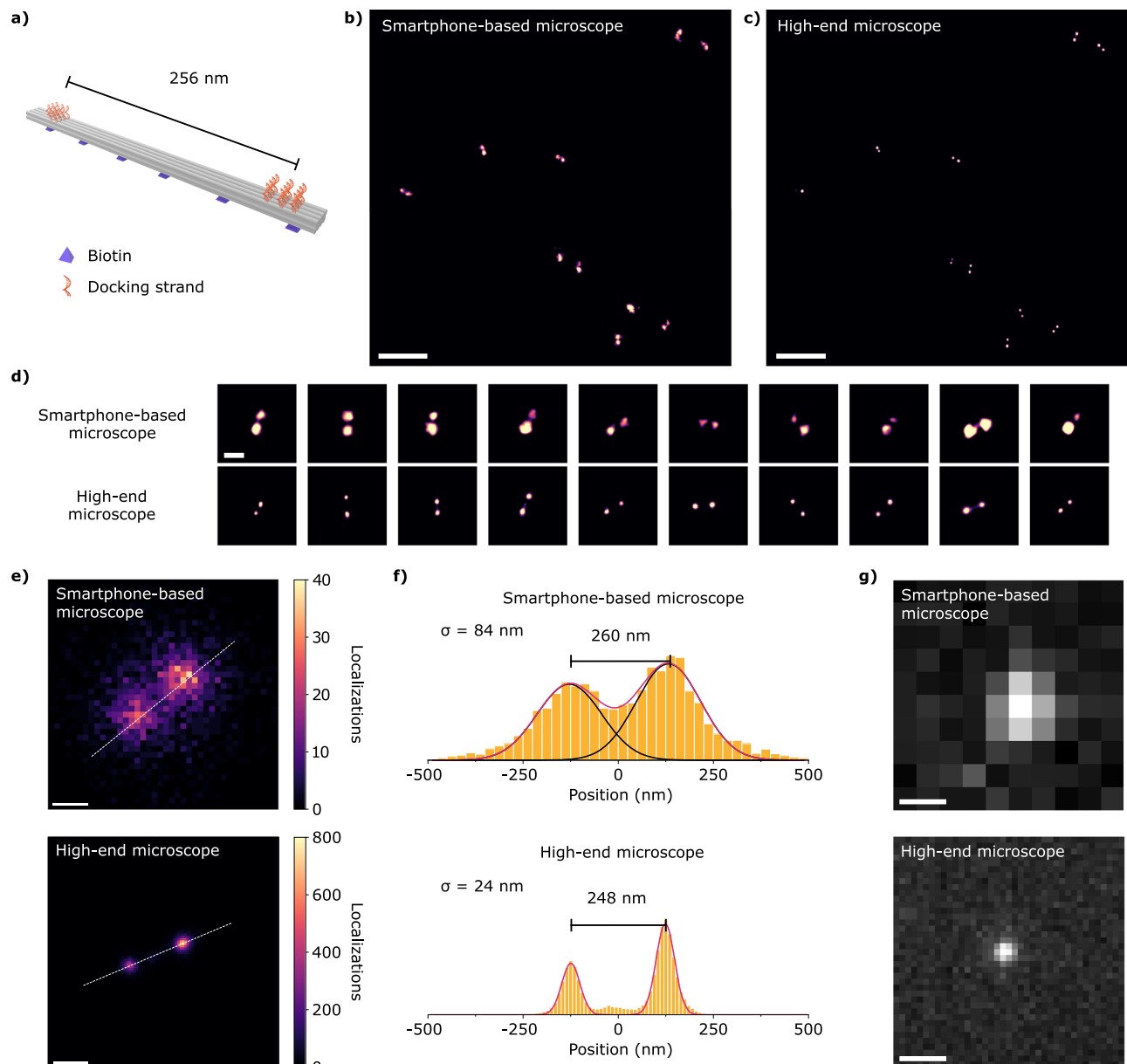

**Fig. 3 | DNA-PAINT with the smartphone-based microscope. a** 8HB DNA origami nanostructure with two docking sites and biotin surface binding functions. Super-resolved images of several nanostructures in an aqueous solution acquired with (**b**) the smartphone-based microscope and (**c**) a high-end microscope. Images are equally scaled. Scale bar: 2 μm. **d** One-to-one comparison of 10 nanostructures. Angle correspondence is observed. Scale bar: 300 nm. **e** Average 2D histograms of localizations of the same set of 10 nanostructures. Bin sizes are 25 nm for the smartphone-based microscope and 10 nm for the high-end microscope. Scale bar: 150 nm. **f** Localization histograms along the central axis (dashed white line) of the nanostructure in (**e**). Bin size: same as in (**e**). **g** Typical diffraction-limited fluorescence spots, i.e., the Point Spread Function of each system, equally scaled. Scale bar: 1 μm. The measurements presented in (**b**) and (**c**) were repeated six times with similar results. Source data are provided as a Source Data file.

analysis[37–39] yielding values of 210 nm and 66 nm for the smartphone-based and the high-end microscopes, respectively (see further details in Supplementary Note 13). These values are in line with visual inspection of individual microtubules (Fig. 4e) and the measurements on DNA origami (Fig. 3).

## Implementation of a bioassay for RNA detection with the smartphone-based microscope

To demonstrate the potential of the smartphone-based microscope for POC measurements, a bioassay was developed and implemented for RNA detection. Inspired by our previous work[40], the assay is based on a DNA origami designed to detect a 30-nucleotide-long Ebola RNA

fragment through DNA-PAINT. Ebola is a highly infectious virus that affects predominantly developing countries with low resources, which would therefore benefit from a reliable and affordable POC detection system[41]. The sensor consists of a ~ 600 nm 8HB dimer constructed upon hybridizing two 8HB monomer structures labeled A-type and B-type origami (see Fig. 5a). TEM images are provided in Supplementary Note 3. The A-type 8HB origami monomer hosts two control DNA-PAINT binding sites on both ends, whereas the B-type monomer hosts one control site and one target site to capture Ebola RNA oligonucleotides.

The working principle of the bioassay is depicted in Fig. 5b. The target Ebola RNA fragment can hybridize along 15 nucleotides

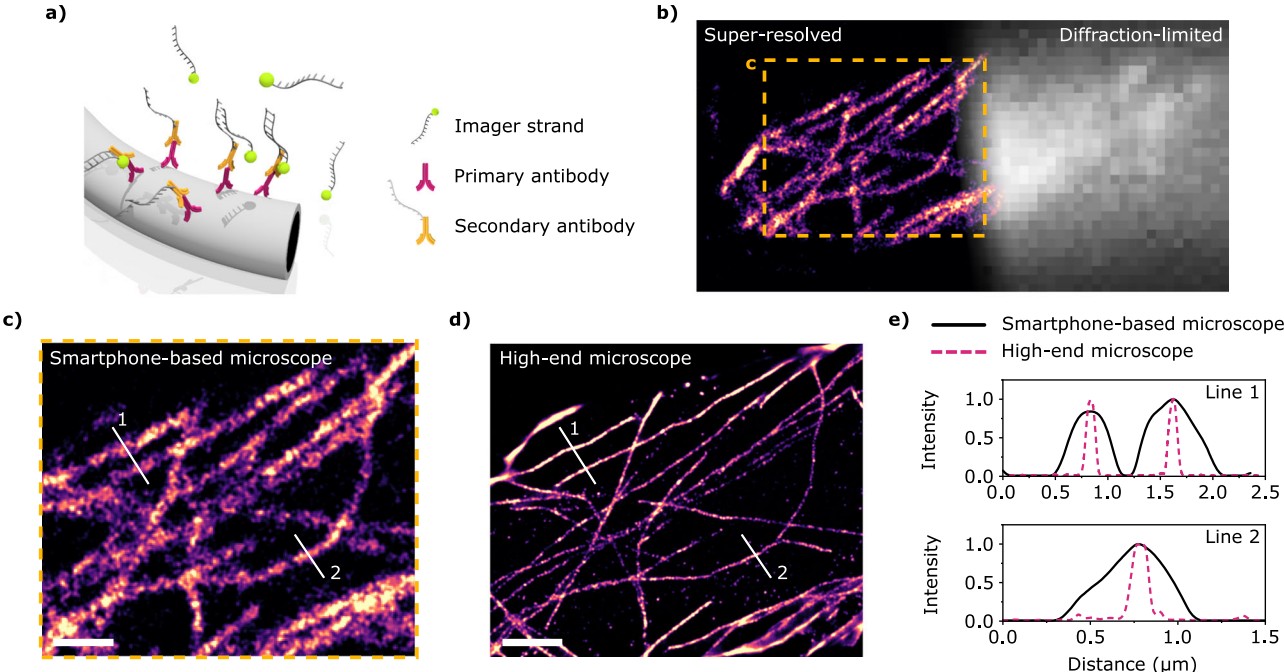

**Fig. 4 | Super-resolution DNA-PAINT imaging on cells. a** Sketch of an immuno-labeled microtubule with primary and secondary antibodies. The secondary antibody is conjugated with a docking DNA strand complementary to the Cy3B-labeled imager strand. **b** Super-resolution and diffraction-limited images of the microtubule network of fixed U2OS cells obtained with the smartphone-based microscope. The super-resolved image represents the localization density, while the diffraction-limited image is the actual fluorescence intensity. Scale bar: 2 μm.
**c**, **d** Comparison of super-resolved images of the microtubule network obtained with the smartphone-based setup (**c**) – a magnified image of the region marked in (**b**) - and with the high-end microscope (**d**). Scale bar: 2 μm. **e** Line profiles across super-resolved microtubules as indicated in (**c**) and (**d**). Black solid lines: smartphone-based microscope. Pink dashed lines: high-end microscope. In total, four different videos were obtained in which the immunolabeled microtubules could be resolved with the smartphone-based microscope. Source data are provided as a Source Data file.

to the complementary DNA anchor strand protruding from the target site. Furthermore, an additional "bridge" strand is employed. A fraction of this bridge strand can hybridize to the remaining 15-base-long segment of the Ebola RNA target. The remaining sequence of the bridge strand is the same as that used in the control sites and thus serves as a docking strand for DNA-PAINT imaging with the fluorogenic imager strand (see Supplementary Note 18 for the staple list and strand sequences). This "sandwich" assay has therefore only two possible outcomes: (i) in the absence of the target RNA, only DNA origami sensors with three sites "on" would be imaged, i.e., the control sites that are always available for DNA-PAINT; and (ii) when the target RNA is present, the DNA origami sensors would now reveal a fourth site "on" indicating that the target RNA has been detected. Figure 5c displays our results, measured with the high-end microscope, that confirm these two possible outcomes for 16 representative DNA origami sensors, 8 for each case (see Methods for sample preparation).

Figure 5d shows the outcomes when the bioassay is implemented with the smartphone-based microscope. The control experiment results in structures with two spots (control column). Expectedly, the two adjacent control binding sites located at the sensor's center are too close to be resolved appropriately by the smartphone-based microscope and are rendered as a larger spot. By contrast, in the presence of the Ebola RNA, the experiment reveals structures with three spots, indicating that the target RNA has been successfully detected (detected column). In assessing the performance of the smartphone-based microscope for detecting RNA fragments using the designed bioassay, we found that 50% of the origami sensors captured the target RNA fragment, whereas this fraction was 66% with the high-end microscope.

## Discussion

We have demonstrated the feasibility of directly detecting single-molecule fluorescence using a simple, low-cost, portable microscope that leverages the image sensors and data handling capacity of mass-produced smartphones. Such ultimate sensitivity was validated in DNA-origami model systems and exploited to demonstrate super-resolution microscopy of cellular microtubules and a proof-of-concept of POC biosensing for RNA detection, both based on single-molecule localization through DNA-PAINT. The setup can be easily paired with various smartphones to detect single molecules, representing a significant leap toward universalizing quantitative sensing. We envision that the smartphone-based microscope presented here will inspire new developments and biotechnological innovations, endowing single-molecule sensitivity to a wide range of applications such as digital bioassays, POC diagnosis, field expeditions, STEM outreach, and life science education.

Several factors have been key to achieving direct detection of single molecules. In contrast to previous works that achieved low background by using LEDs in a TIR configuration[25–28] or a Kretschmann configuration for surface plasmon excitation[24], we used a prism-coupled TIR laser excitation. This configuration allows for highly inclined[42] or TIR illumination[43], thereby providing high irradiance while reducing the background signal (see Methods and Supplementary Note 14). In addition, laser excitation enables a more efficient spectral separation of the fluorescence light using fewer spectral filters. A final factor that leads to a higher SBR is the analysis of raw RGB data, which allowed us to choose the most convenient detection channels. For example, the emission of single ATTO 542 molecules was detected with a 30% higher SBR when the green channel alone was analyzed, compared to the red and green channels combined (Fig. 2e). The increase in SBR is even higher when compared to the total RGB signal.

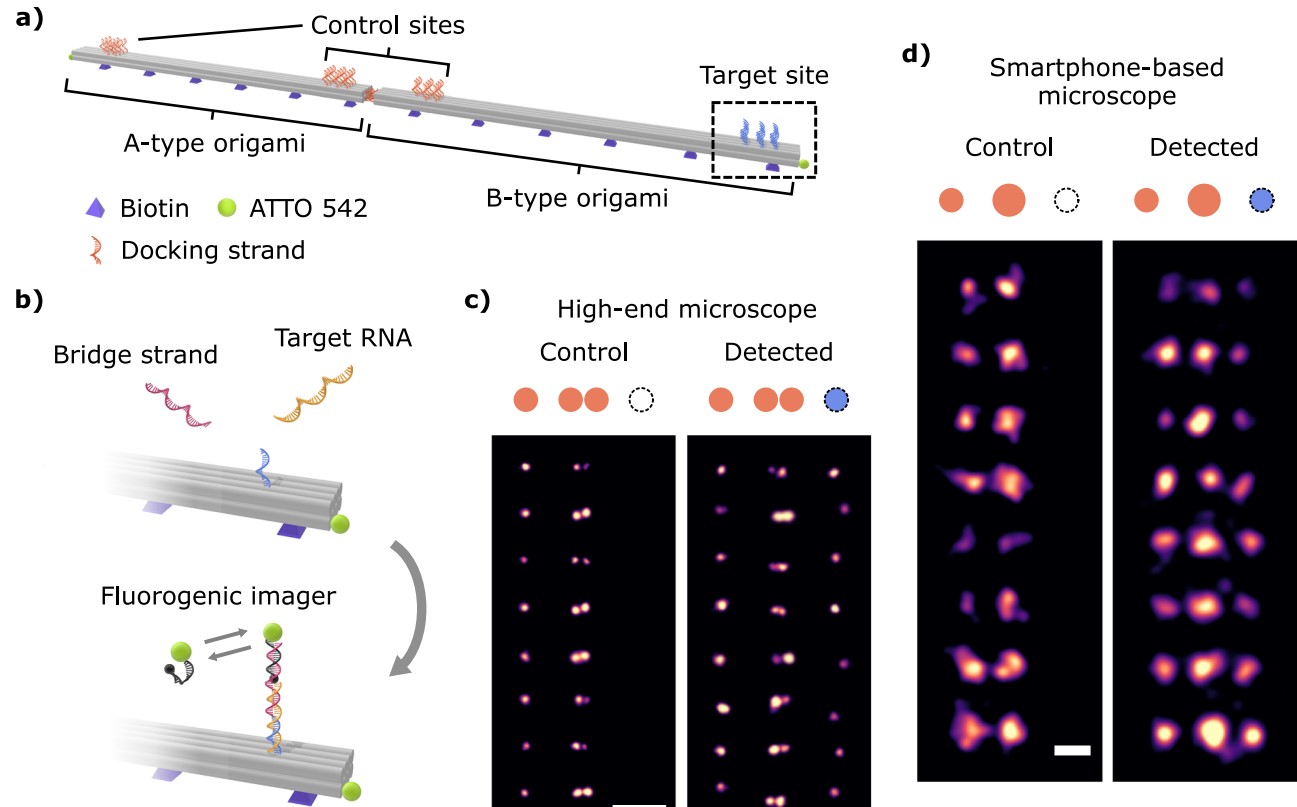

**Fig. 5 | Bioassay for RNA detection using DNA-PAINT imaging. a** Sketch of the DNA origami sensor. An A-type and a B-type 8HB origami monomers assemble to build the sensor with three control sites and one target site. **b** DNA/RNA hybridization principle for RNA detection at the target site (squared dashed box in (**a**)). The protruding DNA strands from the origami partially bind the target RNA. A bridge strand binds to the other half of the RNA fragment. In a DNA-PAINT experiment, the fluorogenic imagers bind to the free nucleotides of the bridge strand and the control sites. **c, d** Eight representative DNA origami sensors displaying the sandwich assay outcome using the high-end and smartphone-based microscopes, respectively. Scale bars: 250 nm. The experiments included in (**c**) and (**d**) were repeated once with comparable results.

Future iterations will focus on improving photon collection efficiency and measurement throughput. This could involve optimizing smartphone camera parameters and software or upgrading optical components at the expense of a small increase in the overall cost. For example, the photon collection efficiency could be improved by upgrading the NA of the objective lens, e.g., using a standard PMMA acrylic Fresnel lens. Using higher-quality interference optical filters in the detection instead of a single, colored glass filters would lead to a higher SBR. Besides upgrading the optics, sample preparation protocols could be optimized to improve signal-to-background ratio and/or reduce acquisition times, for example, by selecting the best-suited combinations of fluorophores and imaging buffers. Also, data processing and analysis times could be reduced by developing specific smartphone apps for single-molecule detection and super-resolution.

With an estimated cost below 350 €, our results bring single-molecule measurements and assays within reach of any research laboratory, biotech company, or higher education institution. The setup requires minimal experience in optics, and a brief tutorial and short training make the microscope accessible to any interested user. Notably, the time-demanding super-resolution imaging measurements yielded comparable results when performed on a standard office desk and an optical table, demonstrating the portable setup's versatility (see Supplementary Note 15 for an exemplary super-resolved image).

Finally, our work paves the way for innovation in personalized assays with single-molecule sensitivity. The low cost and wide availability provide biotechnology enterprises, especially start-ups, with a new dimension to project their applications, leveraging single-molecule sensitivity and the versatility of smartphones. Digital assays based on counting single molecules are easier to validate and calibrate reliably. Massively distributed assays based on smartphones can communicate with remote servers, harnessing the power of big data and AI tools to make single-molecule detection, localization and quantification more efficient and to validate and analyze data. This has the potential to markedly improve the operational efficiency of diagnostic and preventive healthcare processes.

## Methods

This work was carried out according to the Research Integrity Regulations from the SNSF.

### Materials

Acetone: 133-VL54TE, Thommen-Furler. Isopropanol: 172-VL54TE, Thommen-Furler. Quartz coverslips: W9QA0490490170NNNNX1, Microchemicals. Photoresist and developer: AZ 1512 HS and AZ Developer from Microchemicals, respectively. Glass slides: 1000000, Paul Marienfeld. KOH pellets: 1310-58-3, Sigma-Aldrich. 50x TAE: J63931.K3, ThermoFisher. 1 M MgCl₂: J61014.AK, ThermoFisher. 2-component removable glue: Elite Double 16 Fast, Zhermack. NeutrAvidin: 10443985, Fisher Scientific. BSA-biotin: A8549, Sigma-Aldrich. 10x PBS: J75889.K2, ThermoFisher. Double-sided adhesive tape: OCA8146-3, Thorlabs. Fiducial nanoparticles: 60 nm diameter gold nanoparticles, EM. GC60, BBI Solutions. 3,4-Dihydroxybenzoic acid (PCA): 37580-25G-F, Sigma-Aldrich, Protocatechuate 3,4-Dioxygenase from Pseudomonas sp. (PCD): 9029-47-4 P8279-25UN, Sigma-Aldrich. Cyclooctatetraene (COT): 138924-1 G, Sigma-Aldrich. Trolox: 238813-1 G, Sigma-Aldrich. Dimethyl sulfoxide (DMSO): 276855-

100 ML, Sigma-Aldrich. Agarose: 84004, Biozym Scientific GmbH. Amicon Ultra Centrifugal Filter: UFC5100, Merck.

## Buffers preparation

TAE12 buffer was prepared using 1×TAE and 12 mM $MgCl_2$. 1×PPC imaging buffer was prepared using a PCA/PCD oxygen scavenging system and COT as a triplet quencher. In the case of the PPT imaging buffer, we used PCA/PCD plus Trolox as a triplet quencher. PCA aliquots were prepared: 154 mg of PCA in 10 ml of Milli-Q water, and pH was adjusted to 9.0 using NaOH. PCD aliquots were prepared: 8.3 mg of PCD (full bottle) and 11.3 ml of the following buffer (50% glycerol, 50 mM KCl, 1 mM EDTA, and 100 mM Tris). Trolox aliquots were prepared: 100 mg of Trolox, 430 µl of methanol, and 400 µl of NaOH 10 M in 3.2 ml of Milli-Q water. COT aliquots were prepared: 42 mg of COT in 2 ml of DMSO, further diluted in 6 ml of Milli-Q water. All aliquots were stored at −20 °C for up to 6 months. Imaging buffers were prepared using those aliquots and TAE12 buffer to reach final concentrations of 6 mM PCA, 20 nM PCD, and either 2 mM COT for PPC or 2 mM Trolox for PPT.

## DNA origami folding

We used the single-stranded DNA M13mp18 from Bayou Biolabs as a scaffold. All the DNA strands were acquired from Integrated DNA Technologies (IDT) and Biomers GmbH. See Supplementary Notes 17 and 18 for further details on the staples. The DNA origami nanostructures were folded with a 10-fold excess of staples to the scaffold (except for the oligos with fluorophores, which were included with a 1000-fold excess) in a folding buffer, TAE12, with a 20 h non-linear temperature ramp (ramp up to 95 °C, then go to 75 °C 5 min, to 65 °C 20 min, and ramp down 1 °C each 20 min until 20 °C) in a peq-STAR 2X, VWR thermal cycler. Gel electrophoresis was used for purification. The gel contains 0.8% agarose and TAE12 buffer. After adding the gel loading buffer (50% glycerol, 50% TAE12) to the DNA origami solution, the cooled gel was run for 3 h at 70 V. Afterwards, the gel was cut, and the DNA origami solution was extracted via squeezing. The DNA origami concentration was measured with a NanoDrop One$^c$ Spectrophotometer, ThermoFisher Scientific.

For 8HB dimer sensors, the A and B-type monomer solutions were spun down 3 to 4 times using Amicon Ultra centrifugal filters (0.5 ml, 100 kDa MWCO) in 1×TAE 16 mM $MgCl_2$ until the concentration was between 100 and 200 ng/µl. Then, the monomers were mixed in a 1:1 ratio and annealed for 12 h at 42 °C. Finally, the origami sensors were purified by gel electrophoresis using a 1.5% agarose gel, and the same parameters reported above. Again, the extraction was performed via squeezing.

## DNA origami sample preparation

Quartz coverslips were cut into 49 × 16.3 mm substrates with a laser cutter (Q400, Trotec) and left in acetone for 1 h. Next, they were rinsed with isopropanol and soaked for 1 h in fresh isopropanol. Then, the substrates were rinsed with Milli-Q water and dried with compressed nitrogen, followed by 5 min on a hot plate at 100 °C for water desorption. For wide-field/smartphone microscope correlation, 12 nm-thick chromium grids were created over the quartz substrate using photolithography (Direct Writing Laser system: µMLA, Heidelberg Instruments), sputtering (MiniLab 080, Moorfield Nanotechnology), and lift-off.

To remove any resist residue, quartz substrates with grids were cleaned by sonication in acetone for 10 min, followed by sonication in isopropanol for 10 min. Next, the coverslips were rinsed with Milli-Q water and dried with compressed nitrogen (purity 9N5). Then, substrates were exposed to ozone and UV radiation (PSD Pro Serie, Digital UV Ozone System, Novascan) for 10 min. Further, quartz coverslips were placed in a 3 M KOH bath for 10 min. Milli-Q water was used to rinse the quartz substrates, which were then dried with compressed

nitrogen. Finally, substrates were exposed to UV/ozone to make the surface hydrophilic. To make the sandwich chambers, microscope soda lime glass slides were exposed to ozone and UV radiation for 10 min. Then, glass slides were placed in a 3 M KOH bath for 10 min. Milli-Q water was used to rinse the glasses, which were dried with compressed nitrogen. Finally, substrates were exposed to UV/ozone to make the surface hydrophilic.

Once the microscope slides and quartz substrates had been cleaned, 2 stripes of 4 layers of double-sided adhesive tape were applied to the quartz coverslip to form a channel of ~300 µm height. Before closing the chamber, the quartz surface was functionalized with BSA-biotin/NeutrAvidin. First, the surface was incubated for 30 min using 100 µl of 0.5 mg/ml biotinylated BSA diluted in 1xPBS. Second, the surface was rinsed with 200 µl 1×PBS three times, followed by incubation with 100 µl of 0.5 mg/ml NeutrAvidin diluted in 1×PBS:Milli-Q (ratio 19:1) for 30 min. Next, the surface was rinsed three times with 1×PBS and once with TAE12. Next, a droplet of DNA origami solution was incubated to reach the desired surface density with the aid of the fixed dyes incorporated into the DNA origami design (see Supplementary Notes 17 and 18). Typically, a 100 µl solution with 10 pM origami concentration in TAE12 was incubated for 10 to 30 min. Later, the sample was rinsed three times with 200 µl of TAE12. For storage, samples were left in TAE12 buffer, while either PPT or PPC buffer was used for imaging. In the case of super-resolution experiments, fiducial nanoparticles were incubated after DNA origami immobilization, and dye-labeled oligos for DNA-PAINT were added to the imaging buffer. Finally, the chamber was closed with a cleaned glass slide using a 2-component removable glue. Both ends of the channel are glued to prevent oxygen exchange with the environment and buffer evaporation or contamination.

## U2OS cells sample preparation

A human bone osteosarcoma epithelial (U2OS) cell culture with immunolabeled microtubules was purchased from Massive Photonics. The cells were immobilized on a glass coverslip and sealed with an ibidi µ-slide I Luer 0.6 mm height single-channel slide. Immunostaining was done with a primary rat monoclonal antibody against α-tubulin and a secondary monoclonal anti-rat antibody conjugated with docking strands for DNA-PAINT. 90 nm gold nanoparticles were deposited on the surface as fiducial markers for drift correction. In the smartphone-based microscope, the sample was imaged with an imager concentration of 0.5 nM in a 1.5×PPC imaging buffer, in which TAE12 was replaced with 1×PBS. In the high-end microscope, we used an imager concentration of 10 nM in 1×PBS. For storage, the sample was rinsed 3 times with 200 µl of 1×PBS and left at 4 °C.

## Bioassay preparation

After the extraction of the DNA origami sensors, the control sample was prepared in the same way as described in the above subsection *DNA origami sample preparation*. The bioassay to detect the target RNA was performed as follows: a one-pot incubation was done at 4 °C for 7 h. In a LoBind Eppendorf tube, 4 µl of the target RNA at 1 µM were mixed with 4 µl of the bridge strands at 1 µM plus 4 µl of the DNA origami sensors at the stock concentration, typically, around 1 nM. After incubation, the solution was diluted using TAE12 to get 200 µl. Then, the samples were prepared similarly as described in the *DNA origami sample preparation* subsection. On a coverslip surface functionalized with BSA-biotin/NeutrAvidin, a droplet of the diluted bioassay solution is incubated for 30 s (or until the desired surface density is reached). Next, the surface is rinsed three times with TAE12. Then, NPs were added as fiducial markers for drift correction. Then, the imaging buffer solution containing 2.5 nM of the fluorogenic imager was loaded. Finally, the sample was sealed as described before.

## Smartphone-based microscope measurements

Samsung Galaxy S22 Ultra, iPhone 14 Pro, and Huawei P20 Pro smartphones were used to detect a single molecule using our smartphone-based microscope. In the case of the Samsung smartphone, the telephoto camera was used; for the iPhone 14 Pro, the telephoto camera was used; and for the Huawei smartphone, the wide color camera was used. A low-cost colored-glass long-pass filter (10CGA-550, Newport) was used after the inexpensive 0.2 NA, 1.7 mm focal length, fish-eye objective lens (B07FXVJVDP, Richer-R) to collect the fluorescence signal and eliminate the residual scattered laser beam. The prism is a 4 mm diameter N-BK7 half-ball lens (#45-934, Edmund Optics) fixed with optical glue (NOA65, Thorlabs) under a cut microscope slide. Immersion oil (Type F, Leica Microsystems) was placed between the prism and the sample. The laser beam enters the half-ball lens at an incidence angle close to 80° to ensure the entire beam is totally reflected by the coverslip (see Supplementary Note 14). This angle also compensates for the refraction induced by the prism, as the whole laser beam cannot arrive perfectly perpendicular to the surface. The laser module (CW532-020F, Roithner LaserTechnik) has a wavelength of 532 nm and a power of 20 mW. The beam can be focused using the focusing lens incorporated into the laser module. In this work, the laser was focused on the sample plane, rendering an elliptical Gaussian beam with waists $\omega_x = 45.6$ μm and $\omega_y = 7.6$ μm ($1/e^2$ of the peak intensity). Power on the sample plane was measured (S170C with PM100D, Thorlabs) to be $(17.0 \pm 0.5)$ mW, which is equivalent to a top-hat irradiance of 0.76 kW/cm² in an area of 20 × 20 um².

## High-end widefield fluorescence microscope

A schematic of the high-end custom-built wide-field fluorescence microscope is provided in Supplementary Note 16. It is based on an inverted Olympus IX83 body. Lasers with wavelengths of 532 nm and 640 nm (gem 532 and gem 640, Laser Quantum) were used to excite the samples. Each laser line was filtered with a clean-up filter (ZET532/10x, Chroma and ZET642/20x, Chroma) and spatially filtered with a 1x telescope to create Gaussian illumination. The two beams are aligned in a common path with a long-pass filter (RT532rdc, Chroma). Circularly polarized excitation is achieved using a polarizer (LPVISC100-MP2, Thorlabs) and a quarter waveplate (AQWP05M-600, Thorlabs). A neutral-density filter wheel was used to adjust the intensity. The two-color beam was enlarged with a 3.3x beam expander (AC254-030-A-ML and AC508-100-A-ML, Thorlabs) and then focused onto the back focal plane of the objective by a focusing lens (ACT508-300-A-ML, Thorlabs). This system is mounted on a motorized stage (LTM60-50-HSM, OWIS) that can be used to select wide-field, HILO, or TIR illumination conditions. A double-band dichroic mirror (ZT532/640rpc-UF2, Chroma) is used to direct the excitation light to the objective (UPLAPO100xOHR, 1.5 NA, Olympus) and let fluorescence light through. A double-band emission filter (ZET532/640m-TRF, Chroma) is used before the CMOS camera (C14440 ORCA-Fusion, Hamamatsu). The size of the detection PSF, estimated as the width of the intensity profile at $1/e^2$ of its peak, was $\omega = 2\sigma = (0.22 \pm 0.03)$ μm. For single-molecule detection of the ATTO 647 N, we used epi-illumination with a 640 nm Gaussian beam profile with a width ($1/e^2$ of its peak was) of $\omega = (12.6 \pm 0.1)$ μm, while for DNA-PAINT experiments, we used a 532 nm Gaussian beam profile with a width of $\omega = (17.3 \pm 0.1)$ μm in a TIR configuration.

## Recording and processing of videos and pictures with a smartphone

Videos and photos were recorded with MotionCam Pro: RAW Video app version 2.0.2-pro (available at Google Play) when using a Samsung Galaxy S22 Ultra smartphone. In the case of the iPhone 14 Pro, we used the Camera Pro by Moment app version 5.2.5 (available at the Apple store). The FreeDCam app version 4.3.22 was employed for the Huawei

P20 Pro (available at GitHub). Videos acquired with MotionCam were recorded in MCRAW format and exported as a sequence of color (RGB) uncompressed 16-bit DNG images. Videos acquired with Camera Pro were converted to 32-bit TIFF using Adobe Premiere Pro 2024. Videos acquired with FreeDCam were only recorded in RGB MP4 format. All applications can acquire pictures directly in DNG. For DNA-PAINT recordings performed with the Samsung Galaxy S22 Ultra, the active sensor area was cropped using the MotionCam app to 1460 × 1096 px² around its center (40% of its original size, see Supplementary Note 8). As a part of the pre-processing routine, MP4 videos were converted to 8-bit TIFF using Adobe Photoshop, while DNG sequences were converted to TIFF using a custom-made MATLAB code. RGB videos were then split into red, green, and blue channels. Pixel outliers were first removed with Fiji (ImageJ2) software[44]. The pre-processing routine was time-consuming. Downloading the videos to a PC, converting from RAW type to TIFF, and extracting the R, G and B channels involved 1 to 2 h depending on the video file size. A list of video file sizes and formats is given in Supplementary Note 19. Single-molecule intensity traces were extracted and analyzed using custom-made Python codes (see Code availability statement).

The signal-to-background ratio (SBR), or Weber contrast, was calculated from one image as $SBR = (S - B)/B$ where $S$ and $B$ represent the fluorescence and background intensities, respectively. First, the baseline of the smartphone camera was determined from a dark image and subtracted from all pixels. Next, to virtually increase the exposure time, we added the first 5 frames up to get an equivalent exposure time of 1.25 s. Then, for each detected molecule, we defined an inner region of interest (ROI) of 6 × 6 px² and an outer ROI of 8 × 8 px² centered around the diffraction-limited fluorescence spot. We obtained $S$ after averaging the pixel intensities over the inner ROI. In like manner, $B$ was obtained from the average intensity of surrounding pixels. The difference between the outer and inner ROIs produces a 2-pixel-width squared ring from which $B$ was calculated. Spots that showed outer ROIs overlapping other spots were not considered.

The signal-to-noise ratio (SNR) was obtained directly from single-molecule intensity traces, with no frame-averaging, as $SNR = (S - B)/\sigma_S$. Here, $S$ and $\sigma_S$ are the average and the standard deviation of the intensity trace before the molecule photobleaches, and $B$ is the time average of the background signal near the molecule. Traces were obtained from recorded videos after averaging the intensity of a 6-pixel diameter circular ROI around the diffraction-limited spots.

## DNA-PAINT imaging

We performed DNA-PAINT experiments using custom-made sequences for the imager strand and its complementary docking strand. The imager oligonucleotides were conjugated with a Cy3B fluorophore on one end and a fluorescence quencher (BHQ2) on the other, following the general approach described in ref. 34. The imager and docking sequences we used were 5′-BHQ2-AAGTTGTAATGAAGA-Cy3B-3′ and 5′-TTATCTCCTATACAACTTCC-3′, respectively. When hybridized, the 15-nucleotide-long imager strand has a pre-designed mismatch of 3 base pairs, which causes an average temporary bind of $(1.05 \pm 0.04)$ s, as shown in Supplementary Note 11. The imager concentration was 10 nM in 1×PPC imaging buffer for DNA origami imaging. We used an imager concentration of 0.5 nM in 1.5×PPC for the cellular microtubules sample. Picasso software[35] was used to analyze DNA-PAINT videos. Data was fitted using Maximum Likelihood Estimation and further filtered to keep only localizations with a localization precision below 31 nm. For rendering, localizations were un-drifted using nanoparticles as well as origami nanostructures.

## Statistics & reproducibility

Single-molecule statistics were obtained from sample sizes exceeding 100 in all cases. This lower threshold was selected to balance analysis time with the need to ensure high-quality data. No data were excluded

from the analyses. The experiments were not randomized. The Investigators were not blinded to allocation during experiments and outcome assessment.

## Reporting summary

Further information on research design is available in the Nature Portfolio Reporting Summary linked to this article.

## Data availability

The data presented is available in the Zenodo repository: https://doi.org/10.5281/zenodo.13742101. All datasets are publicly accessible without restriction. Source data are provided with this paper.

## Code availability

The code for DNG to TIF conversion with RGB channel extraction, implemented in MATLAB, and the custom Python-based software used to extract intensity traces and to calculate SBR and SNR from videos are available at GitHub[45] (https://github.com/marianobarella/single_molecule_with_a_smartphone).

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

## Acknowledgements

M.L. and G.P.A. thank A. Antonini from SmartMicroOptics and Mathias Lakatos for their helpful advice and fruitful discussions. G.P.A. and F.D.S. would like to thank Alan Szalai, Sebastián Giusti and Lucía Lopez for their support with preliminary samples. M.B. would like to thank Luciano Masullo for the helpful discussions. M.L. acknowledges funding from the University of Fribourg (Fonds de Recherche du Centenaire program). F.D.S. acknowledges the support from Agencia I + D + i, project PICT-2021-01216. G.P.A. acknowledges support from the Swiss National Science Foundation (200021_184687), the National Center of Competence in Research Bio-Inspired Materials NCCR (51NF40_182881), and the Leading House for the Latin American Region through the Research Partnership Grant call 2023 (RPG2302).

## Author contributions

G.P.A. conceived the idea of the project. M.L. and N.F. designed and fabricated the smartphone-based setup. M.L., M.B., F.D.S., and G.P.A. designed the experiments. M.L. and M.B. did the experiments. S.K., K.K., and M.L. designed, synthesized, and characterized the DNA origami nanostructures. M.L. and M.B. analyzed the data and made the figures. M.B., F.D.S., and G.P.A. coordinated the project. M.B., M.L., F.D.S., and G.P.A. wrote the manuscript with input from all the authors.

## Competing interests

The authors declare the following competing interests: M.L., N.F., and G.P.A. have filed a patent application on the described technology to detect single molecules with a smartphone-based fluorescence microscope. The remaining authors have no competing interests.
