## [Transparent Peer Review file · Nature Communications]

Direct single-molecule detection and super-resolution imaging with a low-cost portable smartphone-based microscope

Corresponding Author: Professor Guillermo Acuna

Version 1:

Reviewer comments:

Reviewer #1

(Remarks to the Author)

The authors have presented a well-detailed research work on SMLM (Manuscript ID NCOMMS-24-29133A-Z) with super-resolution imaging on a smartphone platform through successful imaging of DNA origami structures and the microtubule network of U2OS cells. The submitted work looks original and has a scope for further research on the topic. However, the authors need to clarify following points that are critical for the paper :

1. The design of the proposed smartphone microscopic system found to be relatively complex with many mechanical and moving parts, hence, not a user-friendly design as far as the development of the proposed system is concerned in a general laboratory environment.
 2. The imaging of biological specimens such as DNA origami and U2OS cells involves multiple steps of complex sample preparation techniques that again may not be user-friendly in nature. It may involve a steep learning curve for the user to operate the microscope to image such samples. Authors need to justify how this system could be made useful for the user in terms of sample preparation and imaging.
 3. Provide SBR and SNR values of the images of the same samples obtained by both the high-end microscope and the smartphone-based microscope. Compare the values of both devices and comment on the performance of the smartphone-based microscope in terms of these two parameters.
 4. The authors have taken three flagship high-end smartphones as the device for the smartphone microscope. While these smartphones provide good results, only a handful of people have access to such high-end smartphones. The majority of the people use midrange or budget smartphones. Why were these costlier phones have been chosen as a platform instead of midrange smartphones that are widely being used across the world primarily in the developing and least developed nations? Also, how will the performance of the proposed smartphone microscopes will be impacted if midrange smartphones are used ?
 5. How does TIR illumination help in achieving super resolution? Is it due to the use of a laser as a light source instead of an LED? More details and clarification are required in this regard.
 6. Why were single-molecule blinking videos acquired and 3-frame averaging was performed to obtain super-resolution imaging? Why were still microscopic images not taken into account to acquire super-resolution imaging?
 7. How did the authors arrive at the estimation of the PSF waist value and subsequently the gain in resolution from docking site average distance measurements? A proper, detailed calculation is required in this regard.
- Upon receiving the justifiable responses from the authors, the manuscript may be accepted for publications by the journal.

Reviewer #2

(Remarks to the Author)

The authors developed a simplified and miniaturized microscope that uses the cameras of selected (and expensive) smartphones instead of dedicated scientific cameras. The microscope detected single fluorescence dyes in DNA origami structures by analyzing fluorescence fluctuation. Then, the authors achieved super resolution microscopy via DNA PAINT on DNA origami structures and microtubule networks. While the science is sounding and convincing, this work builds upon and extends earlier work by the same authors, and therefore it does not represent yet an important advance. Also, my first

general takeaway is that the authors are framing their work beyond what is offered to the reader. Therefore, I do not believe this work is at the level of what the Nature Communications readership is looking for, and submission to specialized journals is recommended. However, towards improving the quality of the manuscript, I have the following observations:

A main weakness of the manuscript is the lack of clarity in its claims:

1) For example, the authors claim that “our microscope is flexible enough to host any smartphone with a camera,” but results with any smartphone are not shown. Instead, specialized smartphones were used as later indicated by “this type of measurement can be performed with any state-of-the-art smartphone.”

2) “Direct” is used throughout the text multiple times, including in the title, but it is never clarified in the introduction or elsewhere what direct means. One can get the sense that single-molecule detection does not need signal amplification, which is an indirect method, but the connection is never stated. If the authors want to keep using direct, then they should define it and specify “no direct” single molecule detection and super-resolution imaging in the introduction. Otherwise, they must remove such a word. To me, after rereading the manuscript, direct implies that the microscope is ready to perform single molecule detection and DNA-PAINT without the need of adjusting the focal plane, i.e., sample preparation and subsequent sample insertion into the microscope body is just needed to start experiments.

3) In line with the previous issue, the authors also claim in the introduction that “making it compatible with massive personalized applications to be carried out by untrained people”. Untrained is not clear, and a minimum level of training would be expected to be able to use the microscope.

4) There is no mentioning of the drift amount when the microscope is placed on the standard office desk, even though the drift was corrected with nanoparticles and fluorophores. One would expect a microscope of this size and weight to have a large drift, needing an optical table as it was needed in previous work as pointed out in the introduction. However, the connection is not made during result discussions.

5) The authors fail to include the costs of the smartphones in the price list (S2 in the Supplementary File). The smartphones cost over a thousand dollars when released, which will elevate the costs. It is important to be accurate with the cost list because the selected smartphones are crucial for the microscope and the manuscript is framed regarding “cost efficiency” in the smartphone sector. Additional costs for smartphone apps should also be included.

6) The introduction is framed in the context of point-of-care applications enabled by smartphone-based microscope, however, what is demonstrated in the manuscript falls short of that. Rewriting the introduction and clarifying it, or including an application if possible, could help meet the reader expectations. The following review is recommended 10.1016/j.snr.2021.100063.

Other aspects:

7) The manuscript is not clear on the role of the telephoto camera for the Samsung and iPhone in achieving single molecule detection. It is not clear why the color wide camera of the Huawei was used instead of its dedicated Telephoto camera. Perhaps, the Huawei wide camera could help the authors’ claim that any smartphone could be used, but the Huawei results are only briefly shown in the Supplementary Information file. It would be helpful if the authors included a discussion on how smartphone selection may affect single molecule detection and super-resolution.

8) The paragraph starting with Ref 12 by the Ozcan group is misleading. In that paragraph, it is only stated the detection of fluorescent beads. However, in that work, fluorescence beads were added for comparison purposes while the main goal was to image the structure of single molecule DNA, which was stained. Please clarify this.

9) In the same paragraph as previous issue, the sentence “For instance, it would be possible to quantify target molecules under ultra-low analyte concentrations by implementing digital bioassays^{13,14} or to conduct super-resolution fluorescence imaging¹⁵” and references are irrelevant to the flow of the discussion that should be about smartphone-based fluorescence microscopes. A better clarification is needed.

10) Components are listed without considering production costs. For instance, the cost of 3D printed parts and laser cut parts should be included. The authors may use costs from vendor quotes, to get an estimate of how much building the microscope can really cost to other laboratories interested in building the microscope.

11) Price list of relevant components: glass slides should be removed due to the following two reasons. 1) price list is mainly optical components. If the list was not for optical components, then DNA origami staples and cost should be included accordingly. 2) a few glass slides are not sold, but glass boxes.

12) There is no detailed specification on the file sizes recorded using the smartphone apps, and how much space the storage format (for instance, MCRAW) occupies on the phone, particularly for DNA-PAINT raw files. Regular smartphone memory storage is up to 256GB, and data transmission from smartphone to data analysis server is slow. Again, adding memory storage to the smartphone increases the costs of the smartphone.

13) This sentence is unclear “These results are consistent with typical values for the yield of single-stranded DNA labeling with a single dye (see further details in Supplementary Information S4).” Does it mean ssDNA-dye attached to glass? S4

only shows results with DNA origami.

14) Figure S13 caption in the Supplementary Information says "The measurement was performed on a standard office desk for ~3.5 h (see Supplementary Information S13)," pointing out to S13, but I couldn't find S13. Supplementary Information file needs revision.

15) The acronym DNA PAINT is not explained in the introduction.

16) I don't think that stating that data and code are "available upon reasonable request" is sufficient. Shouldn't all data and code be uploaded to a repository, such as the Zenodo data repository?

17) The authors of this manuscript have three additional authors that are not specified in the Zenodo data repository. Please specify author contributions.

Reviewer #3

(Remarks to the Author)

Loretan et al. present a smartphone-based microscope. They test it and show that it can achieve single-molecule detection. Since they can achieve single-molecule detection, they follow up by applying single-molecule localization microscopy with DNA-PAINT, on DNA origami and microtubules.

In recent years, several groups have pushed toward cheaper camera-based setups that achieve single molecule detection, but the one presented here is to my knowledge by far the cheapest to date and the most portable one. Of course, at their price point, they do not compare well with these other solutions in terms of performance, but these are very different categories, and it does not minimize the achievement of performing single molecule detection on a smartphone.

In summary, my main points to improve this manuscript are:

1. The microscope has limitations that can jeopardize potential applications the authors claim, but they are hardly discussed. The main specifications of the microscope, and the recordings, must be clearly presented in the main text.
2. The authors claim that their microscope could be "revolutionary" for applications, but they do not produce one that the microscope would specifically solve. Either the authors do present at least one of these applications or they tune down drastically these claims everywhere needed in the manuscript. Otherwise, these claims are unrealistic and misleading to the reader. The manuscript is interesting as an academic exercise, it does not need to revolutionize the biomedical world.

In more detail, my main concerns:

- Limitations need to be presented clearly.

o The first limitation is the timescale: authors need to clearly state the minimum exposure time required. Figure 2: it is unclear what examples of images are. Is it a single frame from a camera or is it an average over several images? The supplementary material indicates "Single-molecule intensity traces. 16 frames average". What does "traces" refer to, and what is the relation with Fig 2d? Does it mean that the image of Fig 2b was obtained by averaging 16 frames, leading to a 4-second exposure time? This must be stated clearly in the main text. Detecting a single fluorophore that requires several seconds of exposure or only a fraction of a second changes the scope of the study. SBR and SNR must be clearly defined as calculated from a single frame or averages of multiple frames.

In state-of-the-art DNA-PAINT, the exposure time is 100 ms (S Strauss, R Jungmann, Nature Methods, 2020): how does it compare? Can STORM be used or timescales are too slow?

o The authors need to provide the time needed to image the different samples in the main text clearly. Additionally, it would be valuable to give a total time from preparation to localization-generated image. In most applications discussed, time is an important criterion, so this information is valuable in this context.

o What if the field of view? In the raw data used for Fig 3, the field of view provided is $265 \times 177 = 45312$ pixels, from which only a fraction is used (about a fourth) out of the 10 million pixels of the camera. Therefore, it seems that only about 0.1% of the camera chip is used. Authors need to discuss this point. Providing an illumination profile would be very valuable and simple to obtain.

o Can the authors estimate the typical drift in x, y, and z directions when using the microscope? The authors write that a fan is shut down during measurements, but the acquisitions last for a very long time (the DNA origami experiment being close to 3 hours, which is very long in the field). I would expect some extensive heating. The drift in the direction of the objective axis is usually a major issue using a classical TIRF microscope. How do they deal with this? Is the low aperture of the objective sufficient or are phones actively focusing on the sample?

- The authors cite smartphone-based literature, but it would be valuable to refer to attempts at making SMLM cheaper on the microscope end of the problem (for instance, and not limited to, A. Auer et al. ChemPhysChem (2018). 19(22): 3024-3034)) and compare their microscope to these, even if I understand that the approach is different.

Minor concerns

- In the introduction the authors discussed other methods, such as signal amplification. They claim "Moreover, the amplification of the fluorescence signal exhibits an intrinsic dispersion that hinders the implementation in quantitative sensing applications. This variability is further increased by small variations in sample preparation." Can the authors provide a reference?

- Fig 1. Providing traces and analysis for fluorophores imaged with the traditional microscope, to confirm that single molecules were found in the Atto 647N channel would be valuable.

- Fig 2b. There are some horizontal patterns on the microscope image, what is the origin?
- Fig 2c. Why are pixels not visible? This seems like an interpolated image. Can the authors provide a single-frame image without interpolation?
- Fig 2d: Provide a reference for 0 intensity for each trace. Are these traces obtained from single frames or from averaging multiple frames?
- Page 11, the authors write "Thus, among other exciting possibilities, our microscope has the potential to perform SMLM with nanometer resolution". This is a strong overstatement. Current state-of-the-art SMLM microscope, using better than 1.3 NA objectives, optimized illumination, and high-end cameras achieve sub-10nm resolution, and in very extreme cases reach nanometer resolution. It is not reasonable to let the reader believe that a smartphone-based 0.2 NA objective microscope will achieve nanometer resolution in the foreseeable future. Please erase or correct this sentence.
- Page 12, the authors present their imager with Cy3B and a quencher citing Chung et al. Is the probe designed by the authors or a probe designed in another study? If it was designed in another study, please provide a reference and compare it to the known dynamics. Otherwise, please clarify and state that it was prepared for this study.
- In Fig 3, the authors provide "sigma" values that are not defined. The localization precision for DNA-PAINT is routinely estimated using NeNA (Endesfelder et al. 2014), which is directly provided in Picasso, and used by the authors. It would be valuable to provide it. Note that if "sigma" is the standard deviation of a normal distribution, it is by construction smaller than the full width at half maximum usually used on Airy disks to define the resolution of a microscope. Please provide the latter when discussing Fig 3.
- The value for sigma reported for the high-end microscope is, if it is the standard deviation of the normal distribution, not very good for DNA-PAINT. Using their data on Picasso, I estimated the nearest neighbor approach to give a global localization precision of 15 nm, which is again not very good, as one would typically obtain precision in the order of 5 nm. However, the data provided in Fig S9 seem to contradict these numbers, as the localization precision seems very low (average value below 2.5 nm). Can the authors discuss this point?
- What is the time difference between frames (between two exposure periods)? Is there some significant delay or variation? The analysis of dynamical processes is an important application of single-molecule approaches. In such applications, the frame rate needs to be highly precise, and the time difference between frames very low compared to the exposure time, otherwise it will be difficult to analyze such processes.
- Using a similar illumination scheme, the authors claimed in Vietz et al 2019 that the limit of detection on smartphones is 10 fluorophores. The authors give some explanation in the introduction, but it reads as if the previous microscope would have worked better with the current smartphone as it used more traditional optics and lasers. Is it a leap in camera sensor sensitivity in 5 years? Can the authors provide further discussion on this point later in the manuscript?
- The localization on raw images was not performed on smartphones, which seems to be in contradiction with some of the potential applications claimed. Can the author discuss this point?
- In supplementary figures, please specify in the caption when results were obtained with a high-end microscope or with a smartphone-based microscope.

Version 2:

Reviewer comments:

Reviewer #1

(Remarks to the Author)

Authors have addressed all the points that I raised in their revised paper. The submitted work hence can be considered for publication by the journal.

Reviewer #2

(Remarks to the Author)

The authors have done a great job of addressing my comments and the other reviewers' comments. The time the authors dedicated (a year) paid off. I am glad to recommend the manuscript for publication in Nature Communications.

Reviewer #3

(Remarks to the Author)

In this revision, Loretan et al. improved their manuscript, addressing several of my comments. However, I fear that some of my main comments, including the first two, which were the most important, were not addressed. Although the authors claim to agree with these comments, there seems to be what I hope is some misunderstanding.

The authors present a novel method to detect single molecules with a smartphone. It is obvious that all the basic information regarding imaging, including imaging parameters, and providing raw, non-averaged image, should be presented. The absolute minimum would be in figure 2 and presenting this figure. Authors should understand that this request is to help them improve their manuscript. I therefore strongly encourage them to address the remaining comments.

My comments are preceded with the mention "NEW COMMENT".

Reviewer #3 (Remarks to the Author):

Loretan et al. present a smartphone-based microscope. They test it and show that it can achieve single-molecule detection. Since they can achieve single-molecule detection, they follow up by applying single-molecule localization microscopy with DNA-PAINT, on DNA origami and microtubules.

In recent years, several groups have pushed toward cheaper camera-based setups that achieve single molecule detection, but the one presented here is to my knowledge by far the cheapest to date and the most portable one. Of course, at their price point, they do not compare well with these other solutions in terms of performance, but these are very different categories, and it does not minimize the achievement of performing single molecule detection on a smartphone. We thank the Reviewer for the appraisal of our work. We appreciate for bringing up the following points to improve the manuscript.

In summary, my main points to improve this manuscript are:

1. The microscope has limitations that can jeopardize potential applications the authors claim, but they are hardly discussed. The main specifications of the microscope, and the recordings, must be clearly presented in the main text.

We agree with Reviewer 3 in including further details about setup specifications and parameters of the measurements.

Several comments by Reviewer 3 go in this direction. To address them as completely as possible, we have provided the following tables in the revised Supplementary Information:

- Table S2 in Supplementary Information S8 contains the technical specifications of the smartphone-based microscope and the smartphone cameras we have used

- Table S13 in Supplementary Information S19 lists all relevant parameters of each measurement shown in the manuscript

- Table S14 in Supplementary Information S19 provides the frame size and format of the registered data.

We have also extended the discussion to include potential upgrades of the hardware, protocols, and data analysis:

Besides upgrading the optics, sample preparation protocols could be optimized to improve signal-to-background ratio and/or reduce acquisition times, for example, by selecting the best-suited combinations of fluorophores and imaging buffers. Also, data pre-processing and analysis times could be reduced by developing specific smartphone apps for single-molecule detection and super-resolution.

NEW COMMENT: Although the authors “agree” with my main concern that the specifications of the recording “must be clearly presented in the main text”, this is unfortunately not solved, as this does not appear in the main text. They seem to present table S13 as new, but it was already present in the initial submission and did spark my initial main concern. The authors present a manuscript in which they claim to detect single molecule, it is necessary to state in the main text, e.g. the Result part, not the supplementary material, the effective exposure time used to detect a single molecule, and when an image is presented, the reader should clearly understand what is presented. Trying to make sense of what is written in table S13, it seems that for figure 2b, this is 1s/frame (10-frame averaged time series with a single subframe at 100 ms) and for figure 2c 20s/frame (16-frame averaged time series with a single subframe at 331 ms). This is to me an unfair representation of data. Please make the necessary corrections. To report fairly their data, the authors should present at least once a raw data image; this is obviously something a reader would want to see. If the authors still want to present the averaged images, I strongly encourage them to also show single -not averaged- raw images.

2. The authors claim that their microscope could be “revolutionary” for applications, but they do not produce one that the microscope would specifically solve. Either the authors do present at least one of these applications or they tune down drastically these claims everywhere needed in the manuscript. Otherwise, these claims are unrealistic and misleading to the reader. The manuscript is interesting as an academic exercise, it does not need to revolutionize the biomedical world. We agree with Reviewer 3 that using the word revolutionary is unnecessary. We have rephrased the corresponding passages.

NEW COMMENT: I am glad that the authors agree with my concern, but as far as I can read, but there were two occurrences of “revolutionary” in the initial manuscript, there are still two. In my view this as not been corrected. Do not hesitate to clearly write what was modified to clarify.

In more detail, my main concerns:

3. Limitations need to be presented clearly. The first limitation is the timescale: authors need to clearly state the minimum exposure time required. Figure 2: it is unclear what examples of images are. Is it a single frame from a camera or is it an average over several images? The supplementary material indicates “Single-molecule intensity traces. 16 frames average”. What does “traces” refer to, and what is the relation with Fig 2d? Does it mean that the image of Fig 2b was obtained by averaging 16 frames, leading to a 4-second exposure time? This must be stated clearly in the main text. Detecting a single fluorophore that requires several seconds of exposure or only a fraction of a second changes the scope of the study. SBR and SNR must be clearly defined as calculated from a single frame or averages of multiple frames.

We agree with Reviewer 3 that the description of some experimental parameters was not optimal. In Supplementary Information S19, we have improved Table S13 by adding all relevant parameters of each measurement, including whether frame averaging was performed and, in that case, the number of frames averaged.

NEW COMMENT: See above. Please do correct the manuscript in the main text.

4. In state-of-the-art DNA-PAINT, the exposure time is 100 ms (S Strauss, R Jungmann, Nature Methods, 2020): how does it compare? Can STORM be used or timescales are too slow?

In Table S13 of Supplementary Information S19, we provide the exposure time for each DNA-PAINT measurement. The exposure times used for DNA-PAINT with the smartphone setup ranged from 200 to 250 ms. The effective frame time increases with the frame averaging. These integration times are rather long for STORM. Also, the current version of the smartphone-based microscope operates with a single wavelength, whereas for STORM measurements, it is convenient to have a 405 nm laser for photoactivation. Nonetheless, achieving faster integration times would be beneficial in general. In

the revised version, we mention some improvements that could lead to higher SBR and/or shorter integration times (see response to Comment 1 by Reviewer 3).

5. The authors need to provide the time needed to image the different samples in the main text clearly. Additionally, it would be valuable to give a total time from preparation to localization-generated image. In most applications discussed, time is an important criterion, so this information is valuable in this context.

This comment is in line with previous comments by Reviewer 3 and to some extent with comment 2 from Reviewer 1. We consider that this work does not focus on sample preparation, nevertheless, details regarding those procedures can be found in the Methods section. We do agree with the reviewer that it would be beneficial to include more information about the time required to get a localization-based image, including acquisition pre and post processing. To this end, in Supplementary Information S19, we have improved Table S13 to include all relevant parameters of each measurement. We have also included the following sentence:

The total time required to get a super-resolution image was around 5 hours. Data downloading and pre-processing were time-consuming due to the large size of video files (2.8 h recording the video plus 2.2 h of processing, see Methods and Supplementary Information S19 for further details).

6. What if the field of view? In the raw data used for Fig 3, the field of view provided is $265 \times 177 = 45312$ pixels, from which only a fraction is used (about a fourth) out of the 10 million pixels of the camera. Therefore, it seems that only about 0.1% of the camera chip is used. Authors need to discuss this point. Providing an illumination profile would be very valuable and simple to obtain.

The size of the usable Field of View (FOV) of the smartphone-based setup depends on the focal length of the camera used. For the Samsung S22 Ultra, it was $240 \mu\text{m} \times 240 \mu\text{m}$, which corresponds to a few percent of the sensor, 2.2%. In fluorescence mode, the effective FOV was limited by the extent of the excitation beam to an area smaller than the usable FOV for all smartphones. We used a $45.6 \mu\text{m} \times 27.6 \mu\text{m}$ beam size (see Methods section "Smartphone-based microscope measurements"). The values of these parameters were added to the specifications in Table S2 in the Supplementary Information S8.

7. Can the authors estimate the typical drift in x, y, and z directions when using the microscope? The authors write that a fan is shut down during measurements, but the acquisitions last for a very long time (the DNA origami experiment being close to 3 hours, which is very long in the field). I would expect some extensive heating. The drift in the direction of the objective axis is usually a major issue using a classical TIRF microscope. How do they deal with this? Is the low aperture of the objective sufficient or are phones actively focusing on the sample?

We thank Reviewer 3 for raising this important issue. No active drift correction was used in any measurement. Also, the smartphone cameras were used with the focus fixed, i.e., the autofocus function was disabled.

As indicated in the Methods section, lateral sample drift (x, y directions) was measured using 60 nm spherical gold nanoparticles as fiducial markers. In both setups, the lateral drift was of the order of $1 \mu\text{m}$ over tens of minutes. Exemplary lateral drift trajectories are shown in Figure S9 (also see Comment 4 by Reviewer 2), where it can be seen that the high-end microscope drifted less than $1 \mu\text{m}$ in 30 min, and the smartphone setup drifted $\sim 3 \mu\text{m}$ in 94 min. In all cases the amount of drift was manageable and corrected in the post-processing, allowing all the applications shown in the manuscript, including super-resolution imaging by DNA-PAINT.

With respect to the axial drift (z-direction), Reviewer 3 is right that it is critical because it cannot be corrected in post-processing, and he/she is also right in that the low numerical aperture of the objective plays a role in making the influence of drift on the measurements less significant.

As for the usage of the fan, we found out that it contributes to a more stable laser output with reduced intensity fluctuations. However, we found that this is not critical for DNA-PAINT measurements and the fan was switched off.

NEW COMMENT: It is great that the authors could evaluate the magnitude of drift. Now, it would be valuable to have this reported in the manuscript. One needs to know what to expect using such system.

8. The authors cite smartphone-based literature, but it would be valuable to refer to attempts at making SMLM cheaper on the microscope end of the problem (for instance, and not limited to, A. Auer et al. *ChemPhysChem* (2018). 19(22): 3024-3034)) and compare their microscope to these, even if I understand that the approach is different.

We thank Reviewer 3 for this comment, although we believe that making a comparison to other, non-smartphone-based, cost-effective solutions for single-molecule detection would be unfair as it would not consider key aspects of the smartphone-based microscope like portability and connectivity. Nevertheless, in the revised manuscript we have included the following passage with reference to the work by Auer et al. along with another relevant reference:

While other low-cost solutions for single-molecule detection have been developed^{16,17}, the use of smartphone-based setups holds unique potential due to their portability and intrinsic connectivity.

Added references:

16. Auer, A. et al. Nanometer-scale Multiplexed Super-Resolution Imaging with an Economic 3D-DNA-PAINT Microscope. *ChemPhysChem* 19, 3024–3034 (2018).

17. Diederich, B., Then, P., Jügler, A., Förster, R. & Heintzmann, R. cellSTORM—Cost-effective super-resolution on a cellphone using dSTORM. *PLoS One* 14, e0209827 (2019).

Minor concerns

9. In the introduction the authors discussed other methods, such as signal amplification. They claim "Moreover, the

amplification of the fluorescence signal exhibits an intrinsic dispersion that hinders the implementation in quantitative sensing applications. This variability is further increased by small variations in sample preparation.” Can the authors provide a reference?

That sentence refers to the article that is being discussed. It is cited two sentences before (Ref. 19: Trofymchuk, K. et al. Addressable nanoantennas with cleared hotspots for single-molecule detection on a portable smartphone microscope. *Nat Commun* 12, 950 (2021). <https://doi.org/10.1038/s41467-021-21238-9>). We have cited it again at the end of the sentence to avoid any confusion.

10. Fig 1. Providing traces and analysis for fluorophores imaged with the traditional microscope, to confirm that single molecules were found in the Atto 647N channel would be valuable.

We have added the corresponding traces in Supplementary Information S4.

11. Fig 2b. There are some horizontal patterns on the microscope image, what is the origin?

It is due to the slight refractive index mismatch between the objective lens material and the sample substrate.

12. Fig 2c. Why are pixels not visible? This seems like an interpolated image. Can the authors provide a single-frame image without interpolation?

We thank Reviewer 3 for pointing this out. It was an image rendering problem that we fixed in the revised version.

13. Fig 2d: Provide a reference for 0 intensity for each trace. Are these traces obtained from single frames or from averaging multiple frames?

Intensity traces do not present any time average. This is now clearly stated in the main text and in Table S13 of Supplementary Information S19. The intensity after photobleaching corresponds to the background level; the latter is also now clearly stated in the manuscript. Now, Figure 2d caption reads:

The signal after photobleaching represents the background level. No time averaging was performed.

14. Page 11, the authors write “Thus, among other exciting possibilities, our microscope has the potential to perform SMLM with nanometer resolution”. This is a strong overstatement. Current state-of-the-art SMLM microscope, using better than 1.3 NA objectives, optimized illumination, and high-end cameras achieve sub-10nm resolution, and in very extreme cases reach nanometer resolution. It is not reasonable to let the reader believe that a smartphone-based 0.2 NA objective microscope will achieve nanometer resolution in the foreseeable future. Please erase or correct this sentence.

We agree with Reviewer 3. We have changed our statement as follows:

Direct single-molecule fluorescence detection is a prerequisite for imaging beyond the diffraction limit, as with SMLM. Thus, among other exciting possibilities, our microscope has the potential to perform super-resolution microscopy using a mass-consumption smartphone.

15. Page 12, the authors present their imager with Cy3B and a quencher citing Chung et al. Is the probe designed by the authors or a probe designed in another study? If it was designed in another study, please provide a reference and compare it to the known dynamics. Otherwise, please clarify and state that it was prepared for this study.

We used a custom-made sequence following the general approach introduced by Chung et al. We have clarified this in the Methods section:

We performed DNA-PAINT experiments using custom-made sequences for the imager strand and its complementary docking strand. The imager oligonucleotides were conjugated with a Cy3B fluorophore on one end and a fluorescence quencher (BHQ2) on the other, following the general approach described in reference 34. The imager and docking sequences used were 5'-BHQ2-AAGTTGTAATGAAGA-Cy3b-3' and 5'-TTATCTCCTATACAACCTCC-3', respectively.

16. In Fig 3, the authors provide “sigma” values that are not defined. The localization precision for DNA-PAINT is routinely estimated using NeNA (Endesfelder et al. 2014), which is directly provided in Picasso, and used by the authors. It would be valuable to provide it. Note that if “sigma” is the standard deviation of a normal distribution, it is by construction smaller than the full width at half maximum usually used on Airy disks to define the resolution of a microscope. Please provide the latter when discussing Fig 3.

Following the suggestion by Reviewer 3, and Comment 7 by Reviewer 1, we have revised the discussion of the resolution achieved. We refer Reviewer 3 to see our response to Comment 7 of Reviewer 1.

17. The value for sigma reported for the high-end microscope is, if it is the standard deviation of the normal distribution, not very good for DNA-PAINT. Using their data on Picasso, I estimated the nearest neighbor approach to give a global localization precision of 15 nm, which is again not very good, as one would typically obtain precision in the order of 5 nm. However, the data provided in Fig S9 seem to contradict these numbers, as the localization precision seems very low (average value below 2.5 nm). Can the authors discuss this point?

We agree with Reviewer 3 that it might look like there is a contradiction. First, it is important to state that the sigma reported in the main text is, indeed, the localization precision. We refer the Reviewer to our answer to comment 7 of Reviewer 1, which we hope clarifies the usage of the standard deviation (sigma) as the localization precision.

Second, running NeNA on the dataset of Figure 3 acquired with the high-end microscope gives a localization precision of 17.8 nm. At the same time, running NeNA analysis on the data of Figure S9 (now Figure S14) gives a localization precision of 2.5 nm (PPC buffer dataset acquired with the high-end microscope as well). The difference between these two values arises from the used irradiance (see Supplementary Information S19). The latter experiment was performed with a larger irradiance, which increased the intensity signal of the molecules. As the emitted number of photons increased, the localization precision reduced as the localization precision is inversely proportional to the square root of the number of photons [Deschout, H., Zanicchi, F., Mlodzianoski, M. et al. Precisely and accurately localizing single emitters in

fluorescence microscopy. *Nat Methods* 11, 253–266 (2014). <https://doi.org/10.1038/nmeth.2843>].

Last, to support this observation, we show the distribution of photons for both datasets below. Median values are 479 photons for the Figure 3c dataset and 9253 photons for the Figure S14 dataset. The square root of the ratio is 4.4, which partially explains the difference in the obtained localization precision.

18. What is the time difference between frames (between two exposure periods)? Is there some significant delay or variation? The analysis of dynamical processes is an important application of single-molecule approaches. In such applications, the frame rate needs to be highly precise, and the time difference between frames very low compared to the exposure time, otherwise it will be difficult to analyze such processes. We used two settings that led to small “dead time” between frames: sensor cropping and acquisition in RAW mode. In this configuration, no image processing is needed, thus significantly reducing the transfer and storage time. We have measured the difference between the exposure time and the frame time for the Samsung Galaxy S22 Ultra with the mentioned settings. The frame time was 250.1 ms for a 250.0 ms exposure time (0.1 ms difference).

NEW COMMENT: It could be valuable to include this information in the manuscript and explain how this measurement was done.

19. Using a similar illumination scheme, the authors claimed in Vietz et al 2019 that the limit of detection on smartphones is 10 fluorophores. The authors give some explanation in the introduction, but it reads as if the previous microscope would have worked better with the current smartphone as it used more traditional optics and lasers. Is it a leap in camera sensor sensitivity in 5 years? Can the authors provide further discussion on this point later in the manuscript?

We thank Reviewer 3 for raising this point. There are significant differences in the setup used in Vietz et al. and the one presented here. The detection of the signal of 10 fluorophores described in Vietz et al in 2019 was achieved with a system that was neither portable nor affordable. In that work, in addition to using a specialized smartphone that had a black and white camera, a series of research-grade components were used, including a high-quality laser together with research-grade optical components, including mirrors and fluorescence filters. Finally, those experiments were carried out solely on an optical bench.

We have improved the second paragraph of the Discussion section to make these differences with our previous work clearer:

Several factors have been key to achieving direct detection of single molecules. In contrast to previous works that achieved low background by using LEDs in a TIR configuration^{25–28} or a Kretschmann configuration for surface plasmon excitation²⁴, we used a prism-coupled TIR laser excitation. This configuration allows for highly inclined⁴² or TIR illumination⁴³, thereby providing high irradiance while reducing the background signal (see Methods and Supplementary Information S14). In addition, laser excitation enables a more efficient spectral separation of the fluorescence light using fewer spectral filters. A final factor that leads to a higher SBR is the analysis of raw RGB data, which allowed us to choose the most convenient detection channels. For example, the emission of single ATTO 542 molecules was detected with a 30% higher SBR when the green channel alone was analyzed, compared to the red and green channels combined (Figure 2e). The increase in SBR is even higher when compared to the total RGB signal.

20. The localization on raw images was not performed on smartphones, which seems to be in contradiction with some of the potential applications claimed. Can the author discuss this point?

The article focuses on the major achievement of detecting single fluorescent molecules with a portable and affordable setup leveraging the cameras of current smartphones. Other aspects of the computing and connectivity power of smartphones could be taken advantage of in future applications. We agree with Reviewer 3 that it is worth mentioning other potential benefits of the use of smartphones, such as the development of specific applications for single-molecule detection and analysis, including the generation of super-resolution images. To this end, the computing power, connectivity, and coupling to IA machines already present in current smartphones could play crucial roles. We have extended the Discussion to briefly mention these opportunities:

Finally, our work paves the way for innovation in personalized assays with single-molecule sensitivity. The low cost and wide availability provide biotechnology enterprises, especially start-ups, with a new dimension to project their applications, leveraging single-molecule sensitivity and the versatility of smartphones. Digital assays based on counting single molecules are easier to validate and calibrate reliably. Massively distributed assays based on smartphones can communicate with remote servers, harnessing the power of big data and AI tools to make single-molecule detection, localization and quantification more efficient and to validate and analyze data. This could elevate diagnostics and disease prevention to unprecedented levels of efficiency.

21. In supplementary figures, please specify in the caption when results were obtained with a high-end microscope or with a smartphone-based microscope.

We have modified the captions of the respective figures to account for this observation. We have also updated Figures 1a and S1 with photos of the new setup version.

Response to reviewers

We would like to thank all the reviewers for the constructive comments, which have motivated the improvement of our manuscript. We have addressed all the comments and suggestions. Below, we provide a point-by-point response to each one of them (in **bold** font), indicating when the modifications were made to the manuscript (in **blue**).

Reviewer #1 (Remarks to the Author):

The authors have presented a well-detailed research work on SMLM (Manuscript ID NCOMMS-24-29133A-Z) with super-resolution imaging on a smartphone platform through successful imaging of DNA origami structures and the microtubule network of U2OS cells. The submitted work looks original and has a scope for further research on the topic.

We thank the Reviewer for the positive appraisal of our work. Below we provide a point-by-point response to his/her comments.

However, the authors need to clarify following points that are critical for the paper :

1. The design of the proposed smartphone microscopic system found to be relatively complex with many mechanical and moving parts, hence, not a user-friendly design as far as the development of the proposed system is concerned in a general laboratory environment.

We agree with Reviewer 1 that the system has a certain level of complexity. Indeed, its development required substantial efforts to achieve a design that is at the same time capable of detecting single molecules, cost-efficient, and friendly to the end user. Every stage or module of the setup has the degrees of freedom needed to apply it with practically any smartphone and to inspect different positions of the sample. In that sense, the setup is really friendly to the end user. We have confirmed this by letting new users operate it successfully, with their own smartphones, in several demonstrations and hands-on workshops.

In terms of manufacturing, the moving parts and the compactness of the setup involve some complexity. However, compared to high-end microscopes, it can be easily built by personnel of a research laboratory or workshop with standard technical skills.

2. The imaging of biological specimens such as DNA origami and U2OS cells involves multiple steps of complex sample preparation techniques that again may not be user-friendly in nature. It may involve a steep learning curve for the user to operate the microscope to image such samples. Authors need to justify how this system could be made useful for the user in terms of sample preparation and imaging.

We respectfully disagree with Reviewer 1 on this point. Our work has not focused on simplifying sample preparation procedures for DNA origami, biological cells, or any other sample type. This work is about showing that single fluorescent molecules can be detected directly, without the need for signal amplification, with a compact, portable, and user-friendly setup leveraging current smartphone technology.

The variety of samples used was chosen to showcase the remarkable versatility of the new portable microscope to detect single molecules in different contexts, ranging from synthetic nanostructures to biological cells. Protocols for sample preparation are inherent to the showcased techniques, regardless of the microscope being used.

3. Provide SBR and SNR values of the images of the same samples obtained by both the high-end microscope and the smartphone-based microscope. Compare the values of both devices and comment on the performance of the smartphone-based microscope in terms of these two parameters.

We thank Reviewer 1 for the suggestion to include this analysis. We performed the corresponding experiments and compared both microscopes. The results are included and discussed at the end of the “Direct single-molecule detection with the smartphone-based microscope” subsection of the manuscript. Now it reads:

To assess the smartphone-based microscope's performance relative to the high-end microscope, SBR and SNR were estimated for the same fluorophore, ATTO 542, under similar experimental conditions (see Supplementary Information S7 and S19) both on glass and quartz slides. The median values for the high-end microscope were $SBR_{\text{high-end}} = 0.67$ (2.25) and $SNR_{\text{high-end}} = 9.9$ (16.8) on quartz (glass). Notably, on quartz slides, the SBR achieved with the high-end microscope is around 40% lower than the one obtained with the smartphone-based microscope, due to the refractive index mismatch between the high-end objective and the quartz substrate. Conversely, the difference between the SNR values on quartz slides might arise from the low-noise research-grade CMOS camera of the high-end microscope, which reduces the standard deviation of the fluorescence signal, increasing the SNR (see Methods). On glass slides, as expected, both the SBR and SNR recorded on the high-end microscope are higher than the values obtained with smartphone-based counterpart.

4. The authors have taken three flagship high-end smartphones as the device for the smartphone microscope. While these smartphones provide good results, only a handful of people have access to such high-end smartphones. The majority of the people use midrange or budget smartphones. Why were these costlier phones have been chosen as a platform instead of midrange smartphones that are widely being used across the world primarily in the developing and least developed nations? Also, how will the performance of the proposed smartphone microscopes will be impacted if midrange smartphones are used ?

We acknowledge that the smartphones used are probably above average, but they are no longer flagship models. By the time of the submission, the Huawei smartphone was 6 years old and could be considered in the low-price range. The iPhone and Samsung smartphones used were newer, but they had already been 2 years on the market.

Technologies used in smartphones evolve remarkably rapidly and the cameras of all smartphones have been improved continuously. As newer, better-performing cameras are developed for the flagship models of each brand, their former cameras are implemented in lower-range models. In this context, it is expected that in the near future, the great majority of smartphones will deliver a performance comparable to the models used in our work. In the revised version of the manuscript, we have improved the discussion on this issue, making note of the release dates of each smartphone used:

Analogous measurements were performed with an iPhone 14 Pro and a Huawei P20 Pro (see Supplementary Information S6). The smartphone-based setup can detect single-molecule fluorescence with any of those smartphones. The cameras of all employed smartphones are above average, but do not include the latest developments. The Huawei P20 Pro was released in March 2018, the Samsung S22 Ultra in February 2022, and the iPhone 14 in September 2022. Given the current rapid evolution of smartphone technologies, it is expected that in the near future the great majority of smartphones will count with image acquisition systems delivering a performance comparable to the ones used in this work.

5. How does TIR illumination help in achieving super resolution? Is it due to the use of a laser as a light source instead of an LED? More details and clarification are required in this regard.

We thank Reviewer 1 for raising this important point. Indeed, we found that TIR and the use of laser excitation are key aspects to achieve single-molecule sensitivity. TIR illumination effectively reduces the fluorescence background. Using a laser instead of an LED guarantees high irradiance and better spectral decoupling of the excitation light from the fluorescence emission. To further clarify these aspects, we have improved the second paragraph of the Discussion section of the manuscript:

Several factors have been key to achieving direct detection of single molecules. In contrast to previous works that achieved low background by using LEDs in a TIR configuration^{25–28} or a Kretschmann configuration for surface plasmon excitation²⁴, we used a prism-coupled TIR laser excitation. This configuration allows for highly-inclined⁴¹ or TIR illumination⁴², thereby providing high irradiance while reducing the background signal (see Methods and Supplementary Information S14). In addition, using laser excitation enables a more efficient spectral separation of the fluorescence light using fewer spectral filters. A final factor that leads to a higher SBR is the analysis of raw RGB data, which allowed us to choose the most convenient detection channels. For example, the emission of single ATTO 542 molecules was detected with a 30% higher SBR when the green channel alone was analyzed, compared to the red and green channels combined (Figure 2e). The increase in SBR is even higher when compared to the total RGB signal. Analyzing the data in raw mode was also key for precise single-molecule localization.

6. Why were single-molecule blinking videos acquired and 3-frame averaging was performed to obtain super-resolution imaging? Why were still microscopic images not taken into account to acquire super-resolution imaging?

We thank Reviewer 1 for this comment, which allows us to further clarify the treatment of images acquired with the smartphones. We found that averaging three frames delivered better quality signals to be fitted with the single-molecule localization algorithms. We explain this in the main text and illustrate the averaging effect with data in the Supplementary Information:

The detection of single molecules with the smartphone-based setup works at rather low levels of SNR. We found it advantageous to perform a 3-frame average of the video before analysis with Picasso. This procedure improved the symmetry of the detected single-molecule signals, which in turn led to more precise localizations (see Supplementary Information S11).

7. How did the authors arrive at the estimation of the PSF waist value and subsequently the gain in resolution from docking site average distance measurements? A proper, detailed calculation is required in this regard.

To address this comment by Reviewer 1, along with comment 17 by Reviewer 3, we have improved the description of the analysis pipeline used to obtain the single molecule localization precision:

Figure 3f displays the histograms of localizations along the main axis of the average super-resolved nanostructures (Figure 3e). Each histogram was satisfactorily fitted with the sum of two Gaussian functions with, in principle, different amplitudes and positions, but identical standard deviations (width). This allowed us to determine the average distances between docking sites as the distance between the Gaussian's positions, which were 260 nm and 248 nm for the measurements with the smartphone-based microscope and the high-end microscope, respectively. Both distances are in good agreement with the DNA origami design.

A measure of the average localization precision can be estimated as the standard deviation of the Gaussian functions (σ), which were 84 nm for the smartphone-based microscope and 24 nm for the high-end microscope. An estimate of the achieved resolution is provided by the respective full-width at half maximum (FWHM) values of 197 nm and 56 nm ($\text{FWHM} = 2.35\sigma$), which are well below the FWHM of the experimentally determined PSFs: $(1.3 \pm 0.3) \mu\text{m}$ for the smartphone-based setup and $(0.27 \pm 0.01) \mu\text{m}$ for the high-end microscope (see Figure 3g and Supplementary Information S11). The ratio of the FWHMs of the PSF and the super-resolved docking site of the DNA origami provides an estimation of the increase in resolution. These measurements retrieved a resolution increase of 6.6 times for the smartphone-based microscope and 4.8 times for the research-grade microscope. The higher resolution gain observed with the smartphone-based setup is attributable to the greater influence of the DNA-PAINT docking site size on the high-end microscope measurements. Specifically, the 28 nm docking site size is comparable to the 24 nm localization precision of the high-end microscope, whereas it is only one-third of the 84 nm localization precision achieved with the smartphone-based setup. This difference highlights the remarkable performance of the smartphone-based microscope, particularly given that the irradiance on the sample was lower compared to the high-end microscope (see Supplementary Information S19). As an additional characterization, we analyzed the achieved localization precision using the method called NeNa³⁶, based on first-neighbor distances on consecutive frames, on the super-resolved images. Running NeNa on the dataset acquired with the smartphone-based microscope yields a localization precision of 119 nm, a value larger than the standard deviation obtained from the Gaussian fit to the localization histograms (Figure 3f). This is likely due to the fact that NeNa evaluates all localizations in the area analyzed, including non-specific binding events outside the docking sites of the origami, which can contribute to enlarging the average first neighbor distance (see Supplementary Information S10 for further details on localization precision estimation).

Reviewer #2 (Remarks to the Author):

The authors developed a simplified and miniaturized microscope that uses the cameras of selected (and expensive) smartphones instead of dedicated scientific cameras. The microscope detected single fluorescence dyes in DNA origami structures by analyzing fluorescence fluctuation. Then, the authors achieved super resolution microscopy via DNA PAINT on DNA origami structures and microtubule networks. While the science is sounding and convincing, this work builds upon and extends earlier work by the same authors, and therefore it does not represent yet an important advance. Also, my first general takeaway is that the authors are framing their work beyond what is offered to the reader. Therefore, I do not believe this work is at the level of what the Nature Communications readership is looking for, and submission to specialized journals is recommended. However, towards improving the quality of the manuscript, I have the following observations:

We thank the Reviewer for the constructive criticism, though we are convinced that detecting single-fluorescent molecules using a smartphone-based device is a breakthrough for numerous applications. Below, we address all observations made by Reviewer 2 to improve the manuscript.

A main weakness of the manuscript is the lack of clarity in its claims:

1) For example, the authors claim that “our microscope is flexible enough to host any smartphone with a camera,” but results with any smartphone are not shown. Instead, specialized smartphones were used as later indicated by “this type of measurement can be performed with any state-of-the-art smartphone.”

We agree with Reviewer 2 that the mentioned statement was not fully accurate. We meant to refer to the fact that the setup has been designed to host virtually any smartphone with the current standard geometries. Throughout the manuscript we have made changes to address this comment.

With respect to the concern about using specialized smartphones, there is no need to use state-of-the-art smartphones. In the revised version of the manuscript, we have improved the discussion of this issue. We refer Reviewer 2 to our response to point 4 by Reviewer 1.

2) “Direct” is used throughout the text multiple times, including in the title, but it is never clarified in the introduction or elsewhere what direct means. One can get the sense that single-molecule detection does not need signal amplification, which is an indirect method, but the connection is never stated. If the authors want to keep using direct, then they should define it and specify “no direct” single molecule detection and super-resolution imaging in the introduction. Otherwise, they must remove such a word. To me, after rereading the manuscript, direct implies that the microscope is ready to perform single molecule detection and DNA-PAINT without the need of adjusting the focal plane, i.e., sample preparation and subsequent sample insertion into the microscope body is just needed to start experiments.

We use the expression “direct detection of single molecules” to refer to the fact that single-molecule fluorescence is detected directly, without amplifying the fluorescent signal (as in the case of plasmonic sensors) or amplifying the number of emitting

molecules (as in the case of PCR assays). To make this clearer, in the revised version of the manuscript, we explicitly state this in the abstract and the introduction.

Abstract:

We present a novel, low-cost, portable smartphone-based fluorescence microscope capable of detecting single-molecule fluorescence directly, i.e., without the need for any signal amplification.

Introduction:

Here, we present a portable and inexpensive smartphone-based fluorescence microscope capable of detecting single molecules using three commercially available smartphones from different manufacturers. The microscope enables direct single-molecule detection without the need to amplify the fluorescence signal or the number of emitting molecules. These features facilitate the widespread application of single-molecule assays and measurements. Its performance was first demonstrated by detecting single-molecule intensity fluctuations in DNA origami model systems...

3) In line with the previous issue, the authors also claim in the introduction that “making it compatible with massive personalized applications to be carried out by untrained people”. Untrained is not clear, and a minimum level of training would be expected to be able to use the microscope.

We thank Reviewer 2 for raising this point that allowed us to make the manuscript clearer. Indeed, that sentence was imprecise and we have removed it. In the revised version of the manuscript, we refer to the required user training in two passages as follows:

It is a stand-alone unit that includes a laser, a power source, and the necessary optical components in a modular design conceived to optimize sensitivity, portability, affordability, and practicality (additional photographs can be found in Supplementary Information S1). It can be operated following simple user instructions on any conventional table, or even on the floor. Using the microscope requires minimal training, achievable through instructions, a brief tutorial, or a video demonstration.

4) There is no mentioning of the drift amount when the microscope is placed on the standard office desk, even though the drift was corrected with nanoparticles and fluorophores. One would expect a microscope of this size and weight to have a large drift, needing an optical table as it was needed in previous work as pointed out in the introduction. However, the connection is not made during result discussions.

We thank Reviewer 2 for this comment, which allows us to expand the description of the system’s stability. Naturally, the smartphone-based setup presents a larger amount of drift compared to the high-end microscope. Still, the amount of drift is manageable and allows for performing long DNA-PAINT measurements

We note that sample drift varies from measurement to measurement and depends on a series of factors like temperature gradients, and mechanical vibrations.

To illustrate the amount of drift observed in both setups, we have included Figure S10 in the Supplementary Information (and copied below) showing two drift trajectories registered during DNA-PAINT measurements performed with each setup. Figure S10a shows the drift trajectory during the 30-minute-long DNA-PAINT measurement realized with the high-end microscope on an optical table corresponding to the super-resolved image shown in Figure 3c. Figure S10b shows the drift trajectory during the 94-minute-long DNA-PAINT measurement carried out with the smartphone-based microscope on a standard office desk to obtain the super-resolved image shown in Figure 4b.

Figure S10: Average trajectory of the fiducial markers showing sample drift during (a) DNA-PAINT measurements performed with the high-end microscope in an optical table and (b) with the smartphone-based microscope on an office desk. Their corresponding super-resolved images are Figure 3c for (a) and Figure 4b for (b).

In the revised version, we have included a sentence to point this out:

Despite the long recording times, the observed drift with the smartphone-based microscope is manageable even when used on a standard desktop (see Supplementary Information S10 for further details).

5) The authors fail to include the costs of the smartphones in the price list (S2 in the Supplementary File). The smartphones cost over a thousand dollars when released, which will elevate the costs. It is important to be accurate with the cost list because the selected smartphones are crucial for the microscope and the manuscript is framed regarding “cost efficiency” in the smartphone sector. Additional costs for smartphone apps should also be included.

We thank Reviewer 2 for this comment, which we believe is partially in line with comment 4 by Reviewer 1 and comment 1 by Reviewer 2, regarding the requirement of using high-end smartphones. As we explained in the responses to those comments, there is no need to use flagship smartphone models, which typically cost over a thousand dollars. The fact that our setup enables direct single-molecule detection with models between 3 and 6 years old indicates that, in the near future, this capacity will be available in a large fraction of smartphones.

With respect to the price of the smartphones, we note that they vary from country to country and change strongly with time, continuously delivering better performance for a lower price. In fact, the prices of the smartphones we used have reduced significantly since we first submitted this work. Furthermore, the underlying idea is not to get a new smartphone dedicated to the microscope. Rather, it is the microscope that can be used with the already available users' smartphones. For all these reasons, we do not think it is meaningful to include the price of the smartphones, which can be easily retrieved from an internet search for each specific model and local market.

That said, the discussion about the required camera technology and its development in time is opportune. We thank both Reviewers for raising this point, which we think is now properly addressed.

Also, we agree with Reviewer 2 that it is adequate to include the price of the camera app needed, which is now included in the Supplementary Information S2.

6) The introduction is framed in the context of point-of-care applications enabled by smartphone-based microscope, however, what is demonstrated in the manuscript falls short of that. Rewriting the introduction and clarifying it, or including an application if possible, could help meet the reader expectations. The following review is recommended [10.1016/j.snr.2021.100063](https://doi.org/10.1016/j.snr.2021.100063).

We thank Reviewer 2 for making this point. We consider that the main aim of this work is to demonstrate single-molecule fluorescence detection with a portable and affordable smartphone-based microscope. However, we agree with Reviewer 2 that including a proof of principle for a point-of-care application would further illustrate the potential impact of this work. Therefore, we have added a section in which we report on a new set of experiments that showcase the potential of our smartphone based microscope for single-molecule sensing applications. To this end, we developed a bioassay for RNA detection using the DNA origami technique based on our previous work [Kocabey, S., Chiarelli, G., Acuna, G. P., & Ruegg, C. (2023). Ultrasensitive and multiplexed miRNA detection system with DNA-PAINT. *Biosensors and Bioelectronics*, 224, 115053].

We refer the Reviewer to the subsection entitled "Implementation of a bioassay for RNA detection with the smartphone-based microscope" that we have included in the revised version of the manuscript.

Other aspects:

7) The manuscript is not clear on the role of the telephoto camera for the Samsung and iPhone in achieving single molecule detection. It is not clear why the color wide camera of the Huawei was used instead of its dedicated Telephoto camera. Perhaps, the Huawei wide camera could help the authors' claim that any smartphone could be used, but the Huawei results are only briefly shown in the Supplementary Information file. It would be helpful if the authors included a discussion on how smartphone selection may affect single molecule detection and super-resolution.

We thank Reviewer 2 for this comment that allows us to further explain how cameras were selected. We think this is a relevant issue that deserves a deeper discussion because there seems to be an irreversible trend in smartphone technology to include multiple cameras with varying configurations aiming to cover a range of applications.

When we set out to detect single molecules, we identified three camera parameters as crucial: the aperture, focal length (which, together with our microscope, determines the magnification), and size of the sensor pixels. The aperture should be as high as possible (f-number as low as possible) to maximize photon collection efficiency. By contrast, the focal length (magnification) and sensor pixel size should be balanced in an optimum range, which corresponds to the size of the sensor pixel projected on the sample plane of about one-third of the PSF size. This is a well-known issue related to the Nyquist sampling of the single-molecule images (see e.g. Thompson, R. E., Larson, D. R. & Webb, W. W. Precise nanometer localization analysis for individual fluorescent probes. *Biophys. J.* 82 (2002) 2775–2783, or Ober, R. J., Ram, S. & Ward, E. S. Localization accuracy in single-molecule microscopy. *Biophys. J.* 86 (2004) 1185–1200). If the projected pixel size is too large, all the fluorescence signals from the single molecules are focused into one single sensor pixel, which is not optimum because it makes detecting single molecules less efficient and localization less precise. If the projected pixel size is too small, the limited number of fluorescence photons detected from the single molecules is distributed over many sensor pixels, eventually becoming indistinguishable from noise and background. Considering all these factors, we tested all the cameras of the three smartphones that provided a suitable magnification.

In order to clarify this, we have added the following sentences in the main text:

It is important to stress that three camera parameters are crucial for single-molecule detection: the aperture, focal length, and sensor pixel size. A higher aperture (low f-number) is convenient as it provides higher photon collection efficiency. The smartphone camera's focal length and objective lens determine the system's magnification, which in turn defines the size of the single-molecule signals projected on the camera sensor. For optimum sampling of the single-molecule signals, this should be distributed in about 3×3 sensor pixels^{30,31}. This condition limited the choices of suitable cameras for our experiments. In addition, we wanted to have access to RAW data images in order to analyze the data with our algorithms. Considering all these factors, we performed single-molecule experiments with the best-suited camera of each smartphone, namely the wide color camera of the Huawei P20, the telephoto camera of the Samsung Galaxy S22 Ultra, and the telephoto camera of iPhone 14. Further details about the characteristics of each camera are provided in Supplementary Information Table S2.

8) The paragraph starting with Ref 12 by the Ozcan group is misleading. In that paragraph, it is only stated the detection of fluorescent beads. However, in that work, fluorescence beads were added for comparison purposes while the main goal was to image the structure of single molecule DNA, which was stained. Please clarify this.

Reviewer 2 is right. We have corrected that sentence. Now it reads:

Since the pioneering work by the Ozcan group a decade ago¹², which demonstrated the detection of micrometer-long stained double-stranded DNA and fluorescent beads [...]

9) In the same paragraph as previous issue, the sentence “For instance, it would be possible to quantify target molecules under ultra-low analyte concentrations by implementing digital bioassays^{13,14} or to conduct super-resolution fluorescence imaging¹⁵” and references are

irrelevant to the flow of the discussion that should be about smartphone-based fluorescence microscopes. A better clarification is needed.

The references point to articles that describe potential applications, namely single-molecule bioassays and super-resolution. So far, they have not been demonstrated with smartphone-based microscopes. To clarify this, we have modified the sentence as follows:

For instance, it would be possible to quantify target molecules under ultra-low analyte concentrations by implementing digital bioassays^{13,14} or to conduct super-resolution fluorescence imaging¹⁵⁻¹⁷ with a portable and cost-effective smartphone-based instrument.

10) Components are listed without considering production costs. For instance, the cost of 3D printed parts and laser cut parts should be included. The authors may use costs from vendor quotes, to get an estimate of how much building the microscope can really cost to other laboratories interested in building the microscope.

We thank Reviewer 2 for this suggestion. We decided to include rough costs of the components to make clear that the costs of the setup is significantly lower compared to standard microscopes, particularly to other microscopes capable of detecting single molecules. However, it is not the aim of this scientific paper to provide quotes of production. In addition, productions costs will vary significantly among countries.

11) Price list of relevant components: glass slides should be removed due to the following two reasons. 1) price list is mainly optical components. If the list was not for optical components, then DNA origami staples and cost should be included accordingly. 2) a few glass slides are not sold, but glass boxes.

We agree with Reviewer 2. The price of glass slides was removed from the list.

12) There is no detailed specification on the file sizes recorded using the smartphone apps, and how much space the storage format (for instance, MCRAW) occupies on the phone, particularly for DNA-PAINT raw files. Regular smartphone memory storage is up to 256GB, and data transmission from smartphone to data analysis server is slow. Again, adding memory storage to the smartphone increases the costs of the smartphone.

We agree with Reviewer 2 that the data file size is relevant information. Indeed, depending on the application, storing and handling the amount of data recorded could be an issue. The file size of recorded videos depends on the sensor crop size, the data encoding or format, and, of course, the number of frames. For single-molecule intensity traces, the file size of can be recorded on a user smartphone as it will not exceed 1 GB. DNA-PAINT measurement videos are mostly in the range of tens of GB (raw) and do not exceed 200 GB. We have added a table in the Supplementary Information S19 with the sizes required per frame and the number of frames per video of the presented datasets.

We have also added the following line to further explain the data acquisition process:

The total acquisition time for intensity traces is below 5 min plus 15 min of data downloading, pre-processing, and analysis (see Methods and Supplementary Information S19 for further details).

More information on the data acquisition process for the DNA-PAINT measurements is included in our response to comment 5 by reviewer 3.

13) This sentence is unclear “These results are consistent with typical values for the yield of single-stranded DNA labeling with a single dye (see further details in Supplementary Information S4).” Does it mean ssDNA-dye attached to glass? S4 only shows results with DNA origami.

We acknowledge that the sentence mentioned by Reviewer 2 and the explanation in Supplementary Section S4 (now S5) could be improved. The sentence now reads:

These results are consistent with typical reported yields for the incorporation of single-stranded DNA sequences labeled with a single dye during DNA origami self-assembly (see further details in Supplementary Information S5).

And we have improved the explanation in Supplementary Information S4, now S5, as follows:

After counting single molecules that spatially colocalize, we found that 87% of the nanostructures had single ATTO 542 molecules (Figure S6a), 85% had single ATTO 647N molecules (Figure S6b), and 72% had both dyes incorporated (Figure S6c). This result is in agreement with what is expected from the incorporation yield of dyes into a DNA origami, plus some possible contribution from photobleaching of the dyes. For dye incorporation, we followed a reported protocol that can reach 82% efficiency for single dyes¹. If such a maximum yield is reached for both dyes, it is expected that 97% of the origamis have at least one of the two dyes and that 67% of the origamis have both fluorophores. A comparable value was obtained from our observations for the fraction of doubly-labeled origami.

14) Figure S13 caption in the Supplementary Information says “The measurement was performed on a standard office desk for ~3.5 h (see Supplementary Information S13),” pointing out to S13, but I couldn’t find S13. Supplementary Information file needs revision.

Reviewer 2 is right; this reference was incorrect. In the revised version of the Supplementary Information file, the caption of that figure (now Figure S18) points to the Supplementary Information section S19, where the experimental parameters are reported.

15) The acronym DNA PAINT is not explained in the introduction.

The manuscript was modified accordingly. Now it reads:

By implementing on the smartphone platform a type of Single-Molecule Localization Microscopy (SMLM) called DNA-PAINT (DNA-based Points Accumulation for Imaging in Nanoscale Topography), we were able to image both DNA origami structures and microtubule networks of a biological cell.

16) I don't think that stating that data and code are "available upon reasonable request" is sufficient. Shouldn't all data and code be uploaded to a repository, such as the Zenodo data repository?

We have added all data shown in the manuscript to a Zenodo repository and made all the codes for analysis publicly available on a GitHub repository. The code availability statement now reads:

The code for DNG to TIF conversion with RGB channel extraction, implemented in MATLAB, and the custom Python-based software used to extract intensity traces and to calculate SBR and SNR from videos are available at the GitHub repository:

https://github.com/marianobarella/single_molecule_with_a_smartphone

17) The authors of this manuscript have three additional authors that are not specified in the Zenodo data repository. Please specify author contributions.

The Zenodo repository authors list has been updated. The authors' contributions were declared in the Author Contributions statement of the original manuscript.

Reviewer #3 (Remarks to the Author):

Loretan et al. present a smartphone-based microscope. They test it and show that it can achieve single-molecule detection. Since they can achieve single-molecule detection, they follow up by applying single-molecule localization microscopy with DNA-PAINT, on DNA origami and microtubules.

In recent years, several groups have pushed toward cheaper camera-based setups that achieve single molecule detection, but the one presented here is to my knowledge by far the cheapest to date and the most portable one. Of course, at their price point, they do not compare well with these other solutions in terms of performance, but these are very different categories, and it does not minimize the achievement of performing single molecule detection on a smartphone.

We thank the Reviewer for the appraisal of our work. We appreciate for bringing up the following points to improve the manuscript.

In summary, my main points to improve this manuscript are:

1. The microscope has limitations that can jeopardize potential applications the authors claim, but they are hardly discussed. The main specifications of the microscope, and the recordings, must be clearly presented in the main text.

We agree with Reviewer 3 in including further details about setup specifications and parameters of the measurements. Several comments by Reviewer 3 go in this direction. To address them as completely as possible, we have provided the following tables in the revised Supplementary Information:

- **Table S2 in Supplementary Information S8 contains the technical specifications of the smartphone-based microscope and the smartphone cameras we have used**
- **Table S13 in Supplementary Information S19 lists all relevant parameters of each measurement shown in the manuscript**
- **Table S14 in Supplementary Information S19 provides the frame size and format of the registered data.**

We have also extended the discussion to include potential upgrades of the hardware, protocols, and data analysis:

Besides upgrading the optics, sample preparation protocols could be optimized to improve signal-to-background ratio and/or reduce acquisition times, for example, by selecting the best-suited combinations of fluorophores and imaging buffers. Also, data pre-processing and analysis times could be reduced by developing specific smartphone apps for single-molecule detection and super-resolution.

2. The authors claim that their microscope could be “revolutionary” for applications, but they do not produce one that the microscope would specifically solve. Either the authors do present at least one of these applications or they tune down drastically these claims everywhere needed in the manuscript. Otherwise, these claims are unrealistic and misleading to the reader. The manuscript is interesting as an academic exercise, it does not need to revolutionize the biomedical world.

We agree with Reviewer 3 that using the word revolutionary is unnecessary. We have rephrased the corresponding passages.

In more detail, my main concerns:

3. Limitations need to be presented clearly. The first limitation is the timescale: authors need to clearly state the minimum exposure time required. Figure 2: it is unclear what examples of images are. Is it a single frame from a camera or is it an average over several images? The supplementary material indicates “Single-molecule intensity traces. 16 frames average”. What does “traces” refer to, and what is the relation with Fig 2d? Does it mean that the image of Fig 2b was obtained by averaging 16 frames, leading to a 4-second exposure time? This must be stated clearly in the main text. Detecting a single fluorophore that requires several seconds of exposure or only a fraction of a second changes the scope of the study. SBR and SNR must be clearly defined as calculated from a single frame or averages of multiple frames.

We agree with Reviewer 3 that the description of some experimental parameters was not optimal. In Supplementary Information S19, we have improved Table S13 by adding all relevant parameters of each measurement, including whether frame averaging was performed and, in that case, the number of frames averaged.

4. In state-of-the-art DNA-PAINT, the exposure time is 100 ms (S Strauss, R Jungmann, Nature Methods, 2020): how does it compare? Can STORM be used or timescales are too slow?

In Table S13 of Supplementary Information S19, we provide the exposure time for each DNA-PAINT measurement. The exposure times used for DNA-PAINT with the smartphone setup ranged from 200 to 250 ms. The effective frame time increases with the frame averaging. These integration times are rather long for STORM. Also, the current version of the smartphone-based microscope operates with a single wavelength, whereas for STORM measurements, it is convenient to have a 405 nm laser for photoactivation. Nonetheless, achieving faster integration times would be beneficial in general. In the revised version, we mention some improvements that could lead to higher SBR and/or shorter integration times (see response to Comment 1 by Reviewer 3).

5. The authors need to provide the time needed to image the different samples in the main text clearly. Additionally, it would be valuable to give a total time from preparation to localization-generated image. In most applications discussed, time is an important criterion, so this information is valuable in this context.

This comment is in line with previous comments by Reviewer 3 and to some extent with comment 2 from Reviewer 1. We consider that this work does not focus on sample preparation, nevertheless, details regarding those procedures can be found in the Methods section. We do agree with the reviewer that it would be beneficial to include more information about the time required to get a localization-based image, including acquisition pre and post processing. To this end, in Supplementary Information S19, we have improved Table S13 to include all relevant parameters of each measurement. We have also included the following sentence:

The total time required to get a super-resolution image was around 5 hours. Data downloading and pre-processing were time-consuming due to the large size of video files (2.8 h recording the video plus 2.2 h of processing, see Methods and Supplementary Information S19 for further details).

6. What if the field of view? In the raw data used for Fig 3, the field of view provided is $265 \times 177 = 45312$ pixels, from which only a fraction is used (about a fourth) out of the 10 million pixels of the camera. Therefore, it seems that only about 0.1% of the camera chip is used. Authors need to discuss this point. Providing an illumination profile would be very valuable and simple to obtain.

The size of the usable Field of View (FOV) of the smartphone-based setup depends on the focal length of the camera used. For the Samsung S22 Ultra, it was $240 \mu\text{m} \times 240 \mu\text{m}$, which corresponds to a few percent of the sensor, 2.2%. In fluorescence mode, the effective FOV was limited by the extent of the excitation beam to an area smaller than the usable FOV for all smartphones. We used a $45.6 \mu\text{m} \times 27.6 \mu\text{m}$ beam size (see Methods section “Smartphone-based microscope measurements”). The values of these parameters were added to the specifications in Table S2 in the Supplementary Information S8.

7. Can the authors estimate the typical drift in x, y, and z directions when using the microscope? The authors write that a fan is shut down during measurements, but the acquisitions last for a very long time (the DNA origami experiment being close to 3 hours, which is very long in the field). I would expect some extensive heating. The drift in the direction of the objective axis is usually a major issue using a classical TIRF microscope. How do they deal with this? Is the low aperture of the objective sufficient or are phones actively focusing on the sample?

We thank Reviewer 3 for raising this important issue. No active drift correction was used in any measurement. Also, the smartphone cameras were used with the focus fixed, i.e., the autofocus function was disabled.

As indicated in the Methods section, lateral sample drift (x, y directions) was measured using 60 nm spherical gold nanoparticles as fiducial markers. In both setups, the lateral drift was of the order of 1 μm over tens of minutes. Exemplary lateral drift trajectories are shown in Figure S9 (also see Comment 4 by Reviewer 2), where it can be seen that the high-end microscope drifted less than 1 μm in 30 min, and the smartphone setup drifted ~ 3 μm in 94 min. In all cases the amount of drift was manageable and corrected in the post-processing, allowing all the applications shown in the manuscript, including super-resolution imaging by DNA-PAINT.

With respect to the axial drift (z-direction), Reviewer 3 is right that it is critical because it cannot be corrected in post-processing, and he/she is also right in that the low numerical aperture of the objective plays a role in making the influence of drift on the measurements less significant.

As for the usage of the fan, we found out that it contributes to a more stable laser output with reduced intensity fluctuations. However, we found that this is not critical for DNA-PAINT measurements and the fan was switched off.

8. The authors cite smartphone-based literature, but it would be valuable to refer to attempts at making SMLM cheaper on the microscope end of the problem (for instance, and not limited to, A. Auer et al. ChemPhysChem (2018). 19(22): 3024-3034)) and compare their microscope to these, even if I understand that the approach is different.

We thank Reviewer 3 for this comment, although we believe that making a comparison to other, non-smartphone-based, cost-effective solutions for single-molecule detection would be unfair as it would not consider key aspects of the smartphone-based microscope like portability and connectivity. Nevertheless, in the revised manuscript we have included the following passage with reference to the work by Auer et al. along with another relevant reference:

While other low-cost solutions for single-molecule detection have been developed^{16,17}, the use of smartphone-based setups holds unique potential due to their portability and intrinsic connectivity.

Added references:

16. Auer, A. et al. Nanometer-scale Multiplexed Super-Resolution Imaging with an Economic 3D-DNA-PAINT Microscope. ChemPhysChem 19, 3024–3034 (2018).

17. Diederich, B., Then, P., Jügler, A., Förster, R. & Heintzmann, R. cellSTORM—Cost-effective super-resolution on a cellphone using dSTORM. *PLoS One* 14, e0209827 (2019).

Minor concerns

9. In the introduction the authors discussed other methods, such as signal amplification. They claim “Moreover, the amplification of the fluorescence signal exhibits an intrinsic dispersion that hinders the implementation in quantitative sensing applications. This variability is further increased by small variations in sample preparation.” Can the authors provide a reference?

That sentence refers to the article that is being discussed. It is cited two sentences before (Ref. 19: Trofymchuk, K. *et al.* Addressable nanoantennas with cleared hotspots for single-molecule detection on a portable smartphone microscope. *Nat Commun* 12, 950 (2021). <https://doi.org/10.1038/s41467-021-21238-9>). We have cited it again at the end of the sentence to avoid any confusion.

10. Fig 1. Providing traces and analysis for fluorophores imaged with the traditional microscope, to confirm that single molecules were found in the Atto 647N channel would be valuable.

We have added the corresponding traces in Supplementary Information S4.

11. Fig 2b. There are some horizontal patterns on the microscope image, what is the origin?

It is due to the slight refractive index mismatch between the objective lens material and the sample substrate.

12. Fig 2c. Why are pixels not visible? This seems like an interpolated image. Can the authors provide a single-frame image without interpolation?

We thank Reviewer 3 for pointing this out. It was an image rendering problem that we fixed in the revised version.

13. Fig 2d: Provide a reference for 0 intensity for each trace. Are these traces obtained from single frames or from averaging multiple frames?

Intensity traces do not present any time average. This is now clearly stated in the main text and in Table S13 of Supplementary Information S19. The intensity after photobleaching corresponds to the background level; the latter is also now clearly stated in the manuscript. Now, Figure 2d caption reads:

The signal after photobleaching represents the background level. No time averaging was performed.

14. Page 11, the authors write “Thus, among other exciting possibilities, our microscope has the potential to perform SMLM with nanometer resolution”. This is a strong overstatement. Current state-of-the-art SMLM microscope, using better than 1.3 NA objectives, optimized illumination, and high-end cameras achieve sub-10nm resolution, and in very extreme cases reach nanometer resolution. It is not reasonable to let the reader believe that a smartphone-

based 0.2 NA objective microscope will achieve nanometer resolution in the foreseeable future. Please erase or correct this sentence.

We agree with Reviewer 3. We have changed our statement as follows:

Direct single-molecule fluorescence detection is a prerequisite for imaging beyond the diffraction limit, as with SMLM. Thus, among other exciting possibilities, our microscope has the potential to perform super-resolution microscopy using a mass-consumption smartphone.

15. Page 12, the authors present their imager with Cy3B and a quencher citing Chung et al. Is the probe designed by the authors or a probe designed in another study? If it was designed in another study, please provide a reference and compare it to the known dynamics. Otherwise, please clarify and state that it was prepared for this study.

We used a custom-made sequence following the general approach introduced by Chung et al. We have clarified this in the Methods section:

We performed DNA-PAINT experiments using custom-made sequences for the imager strand and its complementary docking strand. The imager oligonucleotides were conjugated with a Cy3B fluorophore on one end and a fluorescence quencher (BHQ2) on the other, following the general approach described in reference 34. The imager and docking sequences used were 5'-BHQ2-AAGTTGTAATGAAGA-Cy3b-3' and 5'-TTATCTCCTATACTTCC-3', respectively.

16. In Fig 3, the authors provide “sigma” values that are not defined. The localization precision for DNA-PAINT is routinely estimated using NeNA (Endesfelder et al. 2014), which is directly provided in Picasso, and used by the authors. It would be valuable to provide it. Note that if “sigma” is the standard deviation of a normal distribution, it is by construction smaller than the full width at half maximum usually used on Airy disks to define the resolution of a microscope. Please provide the latter when discussing Fig 3.

Following the suggestion by Reviewer 3, and Comment 7 by Reviewer 1, we have revised the discussion of the resolution achieved. We refer Reviewer 3 to see our response to Comment 7 of Reviewer 1.

17. The value for sigma reported for the high-end microscope is, if it is the standard deviation of the normal distribution, not very good for DNA-PAINT. Using their data on Picasso, I estimated the nearest neighbor approach to give a global localization precision of 15 nm, which is again not very good, as one would typically obtain precision in the order of 5 nm. However, the data provided in Fig S9 seem to contradict these numbers, as the localization precision seems very low (average value below 2.5 nm). Can the authors discuss this point?

We agree with Reviewer 3 that it might look like there is a contradiction. First, it is important to state that the sigma reported in the main text is, indeed, the localization precision. We refer the Reviewer to our answer to comment 7 of Reviewer 1, which we hope clarifies the usage of the standard deviation (sigma) as the localization precision.

Second, running NeNA on the dataset of Figure 3 acquired with the high-end microscope gives a localization precision of 17.8 nm. At the same time, running NeNA analysis on the data of Figure S9 (now Figure S14) gives a localization precision of 2.5 nm (PPC buffer dataset acquired with the high-end microscope as well). The difference between these two values arises from the used irradiance (see Supplementary Information S19).

The latter experiment was performed with a larger irradiance, which increased the intensity signal of the molecules. As the emitted number of photons increased, the localization precision reduced as the localization precision is inversely proportional to the square root of the number of photons [Deschout, H., Zanicchi, F., Mlodzianoski, M. *et al.* Precisely and accurately localizing single emitters in fluorescence microscopy. *Nat Methods* 11, 253–266 (2014). <https://doi.org/10.1038/nmeth.2843>].

Last, to support this observation, we show the distribution of photons for both datasets below. Median values are 479 photons for the Figure 3c dataset and 9253 photons for the Figure S14 dataset. The square root of the ratio is 4.4, which partially explains the difference in the obtained localization precision.

18. What is the time difference between frames (between two exposure periods)? Is there some significant delay or variation? The analysis of dynamical processes is an important application of single-molecule approaches. In such applications, the frame rate needs to be highly precise, and the time difference between frames very low compared to the exposure time, otherwise it will be difficult to analyze such processes.

We used two settings that led to small “dead time” between frames: sensor cropping and acquisition in RAW mode. In this configuration, no image processing is needed, thus significantly reducing the transfer and storage time. We have measured the difference between the exposure time and the frame time for the Samsung Galaxy S22 Ultra with the mentioned settings. The frame time was 250.1 ms for a 250.0 ms exposure time (0.1 ms difference).

19. Using a similar illumination scheme, the authors claimed in Vietz et al 2019 that the limit of detection on smartphones is 10 fluorophores. The authors give some explanation in the introduction, but it reads as if the previous microscope would have worked better with the current smartphone as it used more traditional optics and lasers. Is it a leap in camera sensor sensitivity in 5 years? Can the authors provide further discussion on this point later in the manuscript?

We thank Reviewer 3 for raising this point. There are significant differences in the setup used in Vietz et al. and the one presented here. The detection of the signal of 10 fluorophores described in Vietz et al in 2019 was achieved with a system that was neither portable nor affordable. In that work, in addition to using a specialized

smartphone that had a black and white camera, a series of research-grade components were used, including a high-quality laser together with research-grade optical components, including mirrors and fluorescence filters. Finally, those experiments were carried out solely on an optical bench.

We have improved the second paragraph of the Discussion section to make these differences with our previous work clearer:

Several factors have been key to achieving direct detection of single molecules. In contrast to previous works that achieved low background by using LEDs in a TIR configuration^{25–28} or a Kretschmann configuration for surface plasmon excitation²⁴, we used a prism-coupled TIR laser excitation. This configuration allows for highly inclined⁴² or TIR illumination⁴³, thereby providing high irradiance while reducing the background signal (see Methods and Supplementary Information S14). In addition, laser excitation enables a more efficient spectral separation of the fluorescence light using fewer spectral filters. A final factor that leads to a higher SBR is the analysis of raw RGB data, which allowed us to choose the most convenient detection channels. For example, the emission of single ATTO 542 molecules was detected with a 30% higher SBR when the green channel alone was analyzed, compared to the red and green channels combined (Figure 2e). The increase in SBR is even higher when compared to the total RGB signal.

20. The localization on raw images was not performed on smartphones, which seems to be in contradiction with some of the potential applications claimed. Can the author discuss this point?

The article focuses on the major achievement of detecting single fluorescent molecules with a portable and affordable setup leveraging the cameras of current smartphones. Other aspects of the computing and connectivity power of smartphones could be taken advantage of in future applications. We agree with Reviewer 3 that it is worth mentioning other potential benefits of the use of smartphones, such as the development of specific applications for single-molecule detection and analysis, including the generation of super-resolution images. To this end, the computing power, connectivity, and coupling to IA machines already present in current smartphones could play crucial roles. We have extended the Discussion to briefly mention these opportunities:

Finally, our work paves the way for innovation in personalized assays with single-molecule sensitivity. The low cost and wide availability provide biotechnology enterprises, especially start-ups, with a new dimension to project their applications, leveraging single-molecule sensitivity and the versatility of smartphones. Digital assays based on counting single molecules are easier to validate and calibrate reliably. Massively distributed assays based on smartphones can communicate with remote servers, harnessing the power of big data and AI tools to make single-molecule detection, localization and quantification more efficient and to validate and analyze data. This could elevate diagnostics and disease prevention to unprecedented levels of efficiency.

21. In supplementary figures, please specify in the caption when results were obtained with a high-end microscope or with a smartphone-based microscope.

**We have modified the captions of the respective figures to account for this observation.
We have also updated Figures 1a and S1 with photos of the new setup version.**

Response to reviewers

We would like to thank all the reviewers for going through the manuscript and our corrections for the second time. We are pleased to hear that they consider our manuscript has improved considerably and that it is predominantly ready for publication. We have addressed all the comments and suggestions. Below, we provide a point-by-point response to Reviewer #3 (in **bold** font), indicating when the modifications were made to the manuscript (in **blue**).

Reviewer #1 (Remarks to the Author):

Authors have addressed all the points that I raised in their revised paper. The submitted work hence can be considered for publication by the journal.

We thank the Reviewer for the positive appraisal of our work.

Reviewer #2 (Remarks to the Author):

The authors have done a great job of addressing my comments and the other reviewers' comments. The time the authors dedicated (a year) paid off. I am glad to recommend the manuscript for publication in Nature Communications.

We thank the Reviewer for the recommendation to publish in Nature Communications.

Reviewer #3 (Remarks to the Author):

In this revision, Loretan et al. improved their manuscript, addressing several of my comments.

We also thank Reviewer #3 for the positive appraisal of our work, and for the new comments to improve the manuscript.

However, I fear that some of my main comments, including the first two, which were the most important, were not addressed. Although the authors claim to agree with these comments, there seems to be what I hope is some misunderstanding.

The authors present a novel method to detect single molecules with a smartphone. It is obvious that all the basic information regarding imaging, including imaging parameters, and providing raw, non-averaged image, should be presented. The absolute minimum would be in figure 2 and presenting this figure. Authors should understand that this request is to help them improve their manuscript. I therefore strongly encourage them to address the remaining comments.

My comments are preceded with the mention "**NEW COMMENT**".

Reviewer #3 (Remarks to the Author):

Loretan et al. present a smartphone-based microscope. They test it and show that it can achieve single-molecule detection. Since they can achieve single-molecule detection, they follow up by applying single-molecule localization microscopy with DNA-PAINT, on DNA origami and microtubules.

In recent years, several groups have pushed toward cheaper camera-based setups that achieve single molecule detection, but the one presented here is to my knowledge by far the cheapest to date and the most portable one. Of course, at their price point, they do not compare well with these other solutions in terms of performance, but these are very different categories, and it does not minimize the achievement of performing single molecule detection on a smartphone.

We thank the Reviewer for the appraisal of our work. We appreciate for bringing up the following points to improve the manuscript.

In summary, my main points to improve this manuscript are:

1. The microscope has limitations that can jeopardize potential applications the authors claim, but they are hardly discussed. The main specifications of the microscope, and the recordings, must be clearly presented in the main text.

We agree with Reviewer 3 in including further details about setup specifications and parameters of the measurements. Several comments by Reviewer 3 go in this direction. To address them as completely as possible, we have provided the following tables in the revised Supplementary Information:

- Table S2 in Supplementary Information S8 contains the technical specifications of the smartphone-based microscope and the smartphone cameras we have used
- Table S13 in Supplementary Information S19 lists all relevant parameters of each measurement shown in the manuscript
- Table S14 in Supplementary Information S19 provides the frame size and format of the registered data.

We have also extended the discussion to include potential upgrades of the hardware, protocols, and data analysis:

Besides upgrading the optics, sample preparation protocols could be optimized to improve signal-to-background ratio and/or reduce acquisition times, for example, by selecting the best-suited combinations of fluorophores and imaging buffers. Also, data pre-processing and analysis times could be reduced by developing specific smartphone apps for single-molecule detection and super-resolution.

NEW COMMENT: Although the authors “agree” with my main concern that the specifications of the recording “must be clearly presented in the main text”, this is unfortunately not solved, as this does not appear in the main text. They seem to present table S13 as new, but it was already present in the initial submission and did spark my initial main concern. The authors

present a manuscript in which they claim to detect single molecule, it is necessary to state in the main text, e.g. the Result part, not the supplementary material, the effective exposure time used to detect a single molecule, and when an image is presented, the reader should clearly understand what is presented. Trying to make sense of what is written in table S13, it seems that for figure 2b, this is 1s/frame (10-frame averaged time series with a single subframe at 100 ms) and for figure 2c 20s/frame (16-frame averaged time series with a single subframe at 331 ms). This is to me an unfair representation of data. Please make the necessary corrections. To report fairly their data, the authors should present at least once a raw data image; this is obviously something a reader would want to see. If the authors still want to present the averaged images, I strongly encourage them to also show single -not averaged- raw images.

We thank Reviewer 3 for this comment, which allowed us to clarify the acquisition parameters of our measurements. This work includes 20 different measurements where a rather large number of different parameters were involved. Thus, we consider it practical to include all parameters in the Supplementary Information clearly referenced throughout the main text. We have improved Table S13, particularly the comments column to clarify the conditions in which each measurement was taken.

Also, we agree with Reviewer 3 that exposure time and total measurement time are key parameters that could be included in the main text. Reviewer 3 is right, the total acquisition time for the image obtained with the high-end microscope shown in Figure 2b was 1 s (0.1 ms x 10). However, for the smartphone-based image shown in Figure 2c, the total acquisition time was 4 s (0.25 ms x 16), not 20 s. In the revised version of the manuscript, we state this more clearly:

“We first imaged the sample using a custom-built high-end widefield fluorescence microscope to detect the ATTO 647N dyes (see Methods). Exciting at 640 nm, we recorded a series of images over time (100 ms exposure time), where the single-step photobleaching of single ATTO 647N molecules could be observed (see Supplementary Note 4 and 19 for experimental details). Figure 2b shows an exemplary image (10-frame average, total acquisition time 1 s), where single ATTO 647N molecules from several DNA origami structures can be clearly detected as a diffraction-limited spot corresponding to the point spread function (PSF) of the system. Next, the sample was moved onto the smartphone-based microscope placed over a standard desk to detect the ATTO 542 molecules excited at 532 nm. The sample position was adjusted to image the region observed previously with the high-end microscope. A Samsung Galaxy S22 Ultra smartphone was used to record a video with the MotionCam Pro app without compression (raw data mode), with an exposure time for each frame of 250 ms. Figure 2c shows a fluorescence image obtained from averaging 16 frames (total acquisition time 4 s – see Supplementary Note 19 for further experimental parameters)”

We also added this type of information for the super-resolution measurements.

“Single-molecule blinking videos on these samples were acquired with both the high-end microscope (100 ms frame time – 18,000 frames) and the smartphone-based setup with the Samsung Galaxy S22 Ultra (250 ms frame time – 40,785 frames), and subsequently analyzed

with Picasso software³⁵ to reconstruct super-resolution images from single-molecule localizations. The detection of single molecules with the smartphone-based setup works at rather low levels of SNR. We found it advantageous to perform a 3-frame average of the video before analysis with Picasso. This procedure improved the symmetry of the detected single-molecule signals, leading to more precise localizations (see Supplementary Note 11), but required the acquisition of 3x more frames. This, in combination with the 2.5x longer frame time, made the total acquisition times with the smartphone-based setup 7.5x longer.”

2. The authors claim that their microscope could be “revolutionary” for applications, but they do not produce one that the microscope would specifically solve. Either the authors do present at least one of these applications or they tune down drastically these claims everywhere needed in the manuscript. Otherwise, these claims are unrealistic and misleading to the reader. The manuscript is interesting as an academic exercise, it does not need to revolutionize the biomedical world.

We agree with Reviewer 3 that using the word revolutionary is unnecessary. We have rephrased the corresponding passages.

NEW COMMENT: I am glad that the authors agree with my concern, but as far as I can read, but there were two occurrences of “revolutionary” in the initial manuscript, there are still two. In my view this has not been corrected. Do not hesitate to clearly write what was modified to clarify.

We do have to apologize for this. This was overseen at our end. In line with the editorial comments, we have rephrased the sentences, including revolutionize and revolutionary. Now they read:

“This development paves the way for innovative applications of massively distributed or personalized assays with single-molecule sensitivity in various fields such as digital bioassays, POC diagnostics, field expeditions, STEM outreach, and life science education.”

“In summary, this work represents a significant advancement towards making single-molecule fluorescence assays and methods widely accessible, with potential applications in various fields, from point-of-care quantitative sensing devices to nanoscale imaging for personalized diagnostics.”

In more detail, my main concerns:

3. Limitations need to be presented clearly. The first limitation is the timescale: authors need to clearly state the minimum exposure time required. Figure 2: it is unclear what examples of images are. Is it a single frame from a camera or is it an average over several images? The supplementary material indicates “Single-molecule intensity traces. 16 frames average”. What does “traces” refer to, and what is the relation with Fig 2d? Does it mean that the image of Fig 2b was obtained by averaging 16 frames, leading to a 4-second exposure time? This must be stated clearly in the main text. Detecting a single fluorophore that requires several seconds of exposure or only a fraction of a second changes the scope of the study. SBR and SNR must be clearly defined as calculated from a single frame or averages of multiple frames.

We agree with Reviewer 3 that the description of some experimental parameters was not optimal. In Supplementary Information S19, we have improved Table S13 by adding all relevant parameters of each measurement, including whether frame averaging was performed and, in that case, the number of frames averaged.

NEW COMMENT: See above. Please do correct the manuscript in the main text.

We thank the reviewer for this comment. We believe this issue has already been addressed in point 1 by adding the effective exposure times to the caption of Figure 2.

4. In state-of-the-art DNA-PAINT, the exposure time is 100 ms (S Strauss, R Jungmann, Nature Methods, 2020): how does it compare? Can STORM be used or timescales are too slow?

In Table S13 of Supplementary Information S19, we provide the exposure time for each DNA-PAINT measurement. The exposure times used for DNA-PAINT with the smartphone setup ranged from 200 to 250 ms. The effective frame time increases with the frame averaging. These integration times are rather long for STORM. Also, the current version of the smartphone-based microscope operates with a single wavelength, whereas for STORM measurements, it is convenient to have a 405 nm laser for photoactivation. Nonetheless, achieving faster integration times would be beneficial in general. In the revised version, we mention some improvements that could lead to higher SBR and/or shorter integration times (see response to Comment 1 by Reviewer 3).

5. The authors need to provide the time needed to image the different samples in the main text clearly. Additionally, it would be valuable to give a total time from preparation to localization-generated image. In most applications discussed, time is an important criterion, so this information is valuable in this context.

This comment is in line with previous comments by Reviewer 3 and to some extent with comment 2 from Reviewer 1. We consider that this work does not focus on sample preparation, nevertheless, details regarding those procedures can be found in the Methods section. We do agree with the reviewer that it would be beneficial to include more information about the time required to get a localization-based image, including acquisition pre and post processing. To this end, in Supplementary Information S19, we have improved Table S13 to include all relevant parameters of each measurement. We have also included the following sentence:

The total time required to get a super-resolution image was around 5 hours. Data downloading and pre-processing were time-consuming due to the large size of video files (2.8 h recording the video plus 2.2 h of processing, see Methods and Supplementary Information S19 for further details).

6. What if the field of view? In the raw data used for Fig 3, the field of view provided is $265 \times 177 = 45312$ pixels, from which only a fraction is used (about a fourth) out of the 10 million pixels of the camera. Therefore, it seems that only about 0.1% of the camera chip is used. Authors need to discuss this point. Providing an illumination profile would be very valuable and simple to obtain.

The size of the usable Field of View (FOV) of the smartphone-based setup depends on the focal length of the camera used. For the Samsung S22 Ultra, it was $240\ \mu\text{m} \times 240\ \mu\text{m}$, which corresponds to a few percent of the sensor, 2.2%. In fluorescence mode, the effective FOV was limited by the extent of the excitation beam to an area smaller than the usable FOV for all smartphones. We used a $45.6\ \mu\text{m} \times 27.6\ \mu\text{m}$ beam size (see Methods section “Smartphone-based microscope measurements”). The values of these parameters were added to the specifications in Table S2 in the Supplementary Information S8.

7. Can the authors estimate the typical drift in x, y, and z directions when using the microscope? The authors write that a fan is shut down during measurements, but the acquisitions last for a very long time (the DNA origami experiment being close to 3 hours, which is very long in the field). I would expect some extensive heating. The drift in the direction of the objective axis is usually a major issue using a classical TIRF microscope. How do they deal with this? Is the low aperture of the objective sufficient or are phones actively focusing on the sample?

We thank Reviewer 3 for raising this important issue. No active drift correction was used in any measurement. Also, the smartphone cameras were used with the focus fixed, i.e., the autofocus function was disabled.

As indicated in the Methods section, lateral sample drift (x, y directions) was measured using 60 nm spherical gold nanoparticles as fiducial markers. In both setups, the lateral drift was of the order of $1\ \mu\text{m}$ over tens of minutes. Exemplary lateral drift trajectories are shown in Figure S9 (also see Comment 4 by Reviewer 2), where it can be seen that the high-end microscope drifted less than $1\ \mu\text{m}$ in 30 min, and the smartphone setup drifted $\sim 3\ \mu\text{m}$ in 94 min. In all cases the amount of drift was manageable and corrected in the post-processing, allowing all the applications shown in the manuscript, including super-resolution imaging by DNA-PAINT.

With respect to the axial drift (z-direction), Reviewer 3 is right that it is critical because it cannot be corrected in post-processing, and he/she is also right in that the low numerical aperture of the objective plays a role in making the influence of drift on the measurements less significant.

As for the usage of the fan, we found out that it contributes to a more stable laser output with reduced intensity fluctuations. However, we found that this is not critical for DNA-PAINT measurements and the fan was switched off.

NEW COMMENT: It is great that the authors could evaluate the magnitude of drift. Now, it would be valuable to have this reported in the manuscript. One needs to know what to expect using such system.

We agree with Reviewer 3. In the revised version of the manuscript, we include the characteristic drift observed:

“...the observed drift with the smartphone-based microscope is manageable even when used on a standard desktop ($\sim 3.5\ \mu\text{m}$ in one hour, see Supplementary Note 10 for further details)...”

8. The authors cite smartphone-based literature, but it would be valuable to refer to attempts at making SMLM cheaper on the microscope end of the problem (for instance, and not limited

to, A. Auer et al. *ChemPhysChem* (2018). 19(22): 3024-3034)) and compare their microscope to these, even if I understand that the approach is different.

We thank Reviewer 3 for this comment, although we believe that making a comparison to other, non-smartphone-based, cost-effective solutions for single-molecule detection would be unfair as it would not consider key aspects of the smartphone-based microscope like portability and connectivity. Nevertheless, in the revised manuscript we have included the following passage with reference to the work by Auer et al. along with another relevant reference:

While other low-cost solutions for single-molecule detection have been developed^{16,17}, the use of smartphone-based setups holds unique potential due to their portability and intrinsic connectivity.

Added references:

16. Auer, A. et al. Nanometer-scale Multiplexed Super-Resolution Imaging with an Economic 3D-DNA-PAINT Microscope. *ChemPhysChem* 19, 3024–3034 (2018).

17. Diederich, B., Then, P., Jügler, A., Förster, R. & Heintzmann, R. cellSTORM—Cost-effective super-resolution on a cellphone using dSTORM. *PLoS One* 14, e0209827 (2019).

Minor concerns

9. In the introduction the authors discussed other methods, such as signal amplification. They claim “Moreover, the amplification of the fluorescence signal exhibits an intrinsic dispersion that hinders the implementation in quantitative sensing applications. This variability is further increased by small variations in sample preparation.” Can the authors provide a reference?

That sentence refers to the article that is being discussed. It is cited two sentences before (Ref. 19: Trofymchuk, K. et al. Addressable nanoantennas with cleared hotspots for single-molecule detection on a portable smartphone microscope. *Nat Commun* 12, 950 (2021). <https://doi.org/10.1038/s41467-021-21238-9>). We have cited it again at the end of the sentence to avoid any confusion.

10. Fig 1. Providing traces and analysis for fluorophores imaged with the traditional microscope, to confirm that single molecules were found in the Atto 647N channel would be valuable.

We have added the corresponding traces in Supplementary Information S4.

11. Fig 2b. There are some horizontal patterns on the microscope image, what is the origin?

It is due to the slight refractive index mismatch between the objective lens material and the sample substrate.

12. Fig 2c. Why are pixels not visible? This seems like an interpolated image. Can the authors provide a single-frame image without interpolation?

We thank Reviewer 3 for pointing this out. It was an image rendering problem that we fixed in the revised version.

13. Fig 2d: Provide a reference for 0 intensity for each trace. Are these traces obtained from single frames or from averaging multiple frames?

Intensity traces do not present any time average. This is now clearly stated in the main text and in Table S13 of Supplementary Information S19. The intensity after photobleaching corresponds to the background level; the latter is also now clearly stated in the manuscript. Now, Figure 2d caption reads:

The signal after photobleaching represents the background level. No time averaging was performed.

14. Page 11, the authors write “Thus, among other exciting possibilities, our microscope has the potential to perform SMLM with nanometer resolution”. This is a strong overstatement. Current state-of-the-art SMLM microscope, using better than 1.3 NA objectives, optimized illumination, and high-end cameras achieve sub-10nm resolution, and in very extreme cases reach nanometer resolution. It is not reasonable to let the reader believe that a smartphone-based 0.2 NA objective microscope will achieve nanometer resolution in the foreseeable future. Please erase or correct this sentence.

We agree with Reviewer 3. We have changed our statement as follows:

Direct single-molecule fluorescence detection is a prerequisite for imaging beyond the diffraction limit, as with SMLM. Thus, among other exciting possibilities, our microscope has the potential to perform super-resolution microscopy using a mass-consumption smartphone.

15. Page 12, the authors present their imager with Cy3B and a quencher citing Chung et al. Is the probe designed by the authors or a probe designed in another study? If it was designed in another study, please provide a reference and compare it to the known dynamics. Otherwise, please clarify and state that it was prepared for this study.

We used a custom-made sequence following the general approach introduced by Chung et al. We have clarified this in the Methods section:

We performed DNA-PAINT experiments using custom-made sequences for the imager strand and its complementary docking strand. The imager oligonucleotides were conjugated with a Cy3B fluorophore on one end and a fluorescence quencher (BHQ2) on the other, following the general approach described in reference 34. The imager and docking sequences used were 5'-BHQ2-AAGTTGTAATGAAGA-Cy3b-3' and 5'-TTATCTCCTATACAACCTTCC-3', respectively.

16. In Fig 3, the authors provide “sigma” values that are not defined. The localization precision for DNA-PAINT is routinely estimated using NeNA (Endesfelder et al. 2014), which is directly provided in Picasso, and used by the authors. It would be valuable to provide it. Note that if “sigma” is the standard deviation of a normal distribution, it is by construction smaller than the full width at half maximum usually used on Airy disks to define the resolution of a microscope. Please provide the latter when discussing Fig 3.

Following the suggestion by Reviewer 3, and Comment 7 by Reviewer 1, we have revised the discussion of the resolution achieved. We refer Reviewer 3 to see our response to Comment 7 of Reviewer 1.

17. The value for sigma reported for the high-end microscope is, if it is the standard deviation of the normal distribution, not very good for DNA-PAINT. Using their data on Picasso, I estimated the nearest neighbor approach to give a global localization precision of 15 nm, which is again not very good, as one would typically obtain precision in the order of 5 nm. However, the data provided in Fig S9 seem to contradict these numbers, as the localization precision seems very low (average value below 2.5 nm). Can the authors discuss this point?

We agree with Reviewer 3 that it might look like there is a contradiction. First, it is important to state that the sigma reported in the main text is, indeed, the localization precision. We refer the Reviewer to our answer to comment 7 of Reviewer 1, which we hope clarifies the usage of the standard deviation (sigma) as the localization precision.

Second, running NeNa on the dataset of Figure 3 acquired with the high-end microscope gives a localization precision of 17.8 nm. At the same time, running NeNA analysis on the data of Figure S9 (now Figure S14) gives a localization precision of 2.5 nm (PPC buffer dataset acquired with the high-end microscope as well). The difference between these two values arises from the used irradiance (see Supplementary Information S19). The latter experiment was performed with a larger irradiance, which increased the intensity signal of the molecules. As the emitted number of photons increased, the localization precision reduced as the localization precision is inversely proportional to the square root of the number of photons [Deschout, H., Zanicchi, F., Mlodzianoski, M. et al. Precisely and accurately localizing single emitters in fluorescence microscopy. *Nat Methods* 11, 253–266 (2014). <https://doi.org/10.1038/nmeth.2843>].

Last, to support this observation, we show the distribution of photons for both datasets below. Median values are 479 photons for the Figure 3c dataset and 9253 photons for the Figure S14 dataset. The square root of the ratio is 4.4, which partially explains the difference in the obtained localization precision.

18. What is the time difference between frames (between two exposure periods)? Is there some significant delay or variation? The analysis of dynamical processes is an important application of single-molecule approaches. In such applications, the frame rate needs to be highly precise, and the time difference between frames very low compared to the exposure time, otherwise it will be difficult to analyze such processes.

We used two settings that led to small “dead time” between frames: sensor cropping and acquisition in RAW mode. In this configuration, no image processing is needed, thus significantly reducing the transfer and storage time. We have measured the difference between the exposure time and the frame time for the Samsung Galaxy S22 Ultra with the mentioned settings. The frame time was 250.1 ms for a 250.0 ms exposure time (0.1 ms difference).

NEW COMMENT: It could be valuable to include this information in the manuscript and explain how this measurement was done.

The influence of the dead time between frames is negligible for the total measurement times of our experiments. Nonetheless, we agree with Reviewer 3 that this is relevant information and include it in Supplementary Note 19, with an explanation of how the dead times were determined:

“The dead time between frames for the Samsung Galaxy S22 Ultra was determined by computing the ratio between total measurement time and the number of frames for a long measurement of 14,554 frames performed under the same conditions reported in the Methods section, particularly, raw acquisition mode and 40% sensor cropping. The ratio corresponds to the total time per frame (frame time), which was determined to be 250.1 ms when we used an exposure time of 250 ms. Thus, the influence of the dead time between frames in the total measurement time of our experiments was negligible, meaning that the frame time can be considered to be the same as the exposure time. For instance, the total measurement time of the image in Figure 2c was 4.0016 seconds, instead of 4 seconds. “

19. Using a similar illumination scheme, the authors claimed in Vietz et al 2019 that the limit of detection on smartphones is 10 fluorophores. The authors give some explanation in the introduction, but it reads as if the previous microscope would have worked better with the current smartphone as it used more traditional optics and lasers. Is it a leap in camera sensor sensitivity in 5 years? Can the authors provide further discussion on this point later in the manuscript?

We thank Reviewer 3 for raising this point. There are significant differences in the setup used in Vietz et al. and the one presented here. The detection of the signal of 10 fluorophores described in Vietz et al in 2019 was achieved with a system that was neither portable nor affordable. In that work, in addition to using a specialized smartphone that had a black and white camera, a series of research-grade components were used, including a high-quality laser together with research-grade optical components, including mirrors and fluorescence filters. Finally, those experiments were carried out solely on an optical bench.

We have improved the second paragraph of the Discussion section to make these differences with our previous work clearer:

Several factors have been key to achieving direct detection of single molecules. In contrast to previous works that achieved low background by using LEDs in a TIR configuration^{25–28} or a Kretschmann configuration for surface plasmon excitation²⁴, we used a prism-coupled TIR laser excitation. This configuration allows for highly inclined⁴² or TIR illumination⁴³, thereby providing high irradiance while reducing the background signal (see Methods and Supplementary Information S14). In addition, laser excitation enables a more efficient spectral separation of the fluorescence light using fewer spectral filters. A final factor that leads to a higher SBR is the analysis of raw RGB data, which allowed us to choose the most convenient detection channels. For example, the emission of single ATTO 542 molecules was detected with a 30% higher SBR when the green channel alone was analyzed, compared to the red and green channels combined (Figure 2e). The increase in SBR is even higher when compared to the total RGB signal.

20. The localization on raw images was not performed on smartphones, which seems to be in contradiction with some of the potential applications claimed. Can the author discuss this point?

The article focuses on the major achievement of detecting single fluorescent molecules with a portable and affordable setup leveraging the cameras of current smartphones. Other aspects

of the computing and connectivity power of smartphones could be taken advantage of in future applications. We agree with Reviewer 3 that it is worth mentioning other potential benefits of the use of smartphones, such as the development of specific applications for single-molecule detection and analysis, including the generation of super-resolution images. To this end, the computing power, connectivity, and coupling to IA machines already present in current smartphones could play crucial roles. We have extended the Discussion to briefly mention these opportunities:

Finally, our work paves the way for innovation in personalized assays with single-molecule sensitivity. The low cost and wide availability provide biotechnology enterprises, especially start-ups, with a new dimension to project their applications, leveraging single-molecule sensitivity and the versatility of smartphones. Digital assays based on counting single molecules are easier to validate and calibrate reliably. Massively distributed assays based on smartphones can communicate with remote servers, harnessing the power of big data and AI tools to make single-molecule detection, localization and quantification more efficient and to validate and analyze data. This could elevate diagnostics and disease prevention to unprecedented levels of efficiency.

21. In supplementary figures, please specify in the caption when results were obtained with a high-end microscope or with a smartphone-based microscope.

We have modified the captions of the respective figures to account for this observation. We have also updated Figures 1a and S1 with photos of the new setup version.